# Learning to Taste 🍷 : A Multimodal Wine Dataset

**Thoranna Bender** 🍇   **Simon Moe Sørensen** 🍇   **Alireza Kashani** 🍷   **K. Eldjarn Hjorleifsson** 🍃
**Grethe Hyldig** 🍇   **Søren Hauberg** 🍇   **Serge Belongie** 🥂   **Frederik Warburg** 🍇

🍇 Technical University of Denmark  🍷 Vivino
🍃 California Institute of Technology  🥂 University of Copenhagen

## Abstract

We present *WineSensed*, a large multimodal wine dataset for studying the relations between visual perception, language, and flavor. The dataset encompasses 897k images of wine labels and 824k reviews of wines curated from the Vivino platform. It has over 350k unique bottlings, annotated with year, region, rating, alcohol percentage, price, and grape composition. We obtained fine-grained flavor annotations on a subset by conducting a wine-tasting experiment with 256 participants who were asked to rank wines based on their similarity in flavor, resulting in more than 5k pairwise flavor distances. We propose a low-dimensional concept embedding algorithm that combines human experience with automatic machine similarity kernels. We demonstrate that this shared concept embedding space improves upon separate embedding spaces for coarse flavor classification (alcohol percentage, country, grape, price, rating) and aligns with the intricate human perception of flavor.

## 1   Introduction

Vision, language, audio, touch, smell, and taste are sensory inputs that ground humans in a shared representation, which enables us to interact, converse, and create. Recent advances in multimodal learning have shown that combining diverse modalities in a shared representation leads to useful and better-grounded models [Girdhar et al., 2023, Chen et al., 2023]. Inspired by recent progress, we propose to add flavor to the list of modalities used to learn shared representations.

As a first step towards modeling flavor, we focus on wine since (1) wines have been studied for centuries, (2) their flavors have been carefully categorized, and (3) classification systems exist to ensure that flavor is near-consistent across bottles of the same unique bottling.

We bridge the gap between the machine learning and food science communities by presenting WineSensed, a multimodal wine dataset that consists of images, user reviews, and flavor annotations. Our motivation is twofold. On one hand, internet photos and user reviews are a scalable source of data, offering abundant, diverse, and easily accessible insights into wine qualities. On the other hand, human flavor annotations, while not as scalable, provide a more direct and granular understanding of the wines' flavor profile. By combining these resources, we aim to capture the best of both worlds, yielding a richer, more intricate dataset.

We organized a large sensory study to obtain human-annotated flavor profiles of the wines. The study applies the "Napping" methodology [Pagès, 2005], which is commonly used to conduct consumer surveys [Kim et al., 2013, Ribeiro et al., 2020]. In this study, 256 participants annotated their perceived taste similarities of various wines. In Fig. 1, the "human kernel" illustrates how participants

Submitted to the 37th Conference on Neural Information Processing Systems (NeurIPS 2023) Track on Datasets and Benchmarks. Do not distribute.

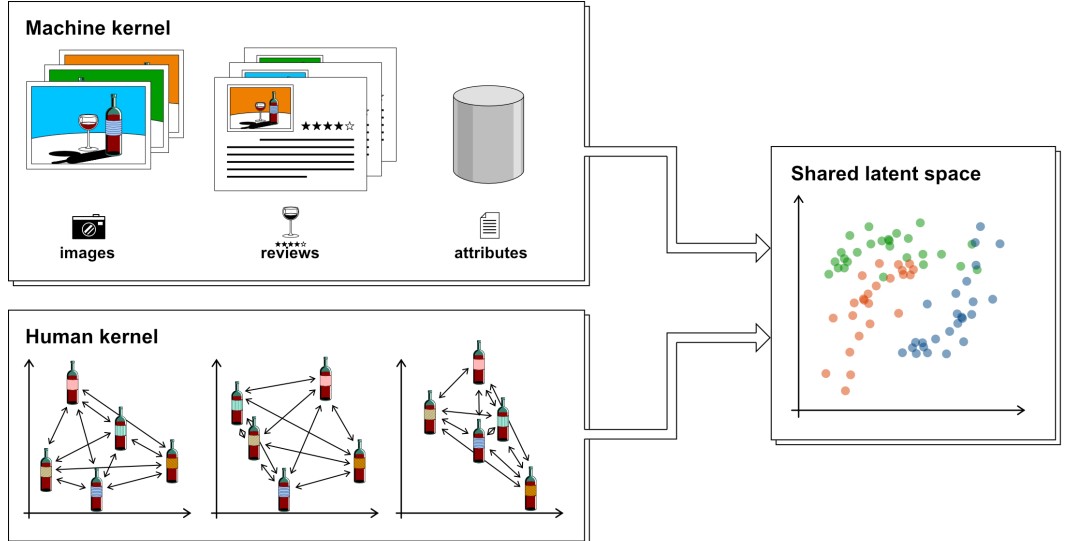

Figure 1: **Flavor as an additional data modality.** The WineSensed dataset consists of a large collection of images, user reviews, and metadata about unique bottlings (upper left). In a large user study, we collected flavor annotations of over 100 wines using the "Napping" method [Pagès, 2005], where participants were asked to place wines on a sheet of paper based on their perceived taste similarity (lower left). We propose an algorithm to combine these data modalities into a shared representation (right) and find that using taste annotations as an additional modality improves performance in downstream tasks.

were instructed to place wines on a sheet of paper based on how similar they perceived their flavor to be. The Napping method enabled us to annotate wine flavors with a high level of detail and harness the perception of a broad spectrum of individuals. It scales well, as asking a participant to annotate five wines yields 10 pairwise annotations. All participants combined annotated more than 5k flavor distances.

To complement these annotations, we curate images of wine labels, user reviews, and wine attributes (country of origin, alcohol percentage, price, and grape composition) from the Vivino platform, a popular online social network for wine enthusiasts.[1] WineSensed, therefore, represents a large, multimodal dataset that merges user-generated content with sensory assessments, bridging the gap between subjective consumer perception and objective flavor profiles.

Along with the dataset, we propose *Flavor Embeddings from Annotated Similarity & Text-Image* (FEAST) that leverages recent developments in large multimodal models to embed user reviews and images of wine labels into a low-dimensional, latent representation that contains semantic and structural information that correlates with taste. Our model aligns this representation with the flavor annotations from our user study. We find that this combined representation yields a "flavor space" that models coarse flavor concepts like alcohol percentage, country, grape, and the year of production, while also being aligned with more intricate human perception of flavor.

Experimentally, we find (1) that using the pairwise distances (rather than ordering) of the annotated wines improves the flavor representation, which confirms the established methodology in food science, and validates our annotation process. (2) We discover that using multiple data modalities (images, text, and flavor annotations) boosts the flavor representations, highlighting the usefulness of our multimodal dataset. (3) Finally, we show that the proposed multimodal model produces a flavor space with a high alignment with humans' perception of flavor.

---

[1] https://vivino.com

## 2 Background and related work

**Multimodal representations.** Learning a shared representation between modalities can reveal useful representations that generalize well and appear grounded in reality. Pioneering work [de Sa, 1994] proposes to learn the correlation between vision and audio. A number of deep learning methods propose to use large collections of weakly annotated data to learn shared vision-language representations [Joulin et al., 2016, Desai and Johnson, 2021, Radford et al., 2021b, Mahajan et al., 2018], shared audio-text representations [Agostinelli et al., 2023], shared vision-audio representations [Ngiam et al., 2011, Owens et al., 2016, Arandjelovic and Zisserman, 2017, Narasimhan et al., 2022, Hu et al., 2022], shared vision-touch [Yang et al., 2022] representations, or shared sound and Inertial Measurement Unit (IMU) representations [Chen et al., 2023]. Recently, ImageBind [Girdhar et al., 2023] showed that images can bind multiple modalities (images, text, audio, depth, thermal, IMU) into a shared representation. While recent advances in other areas of multimodal learning have been fueled by large datasets, the difficulty of quantifying and collecting high-quality flavor data has made it challenging for the machine learning community to develop similar representations for flavor.

**Quantifying flavor.** Understanding and engineering *flavor* is a central part of food science and essential in the quest towards healthy and sustainable food production [Savage, 2012], but the use of machine learning methods to this end is still in its infancy. Fuentes et al. [2019] found a correlation between seasonal weather characteristics, and wine quality and aroma profiles, thereby verifying what wine producers have long held to be true. Similarly, Gupta [2018] found that sulfur dioxide, pH, and alcohol levels are useful for predicting wine quality. Due to the difficulty of gathering quality perception data, much work focuses on how 'low-level' chemical aspects related to 'high-level' taste properties, e.g. in assessing the quality of chocolate and beer [Gunaratne et al., 2019, Gonzalez Viejo et al., 2018].

Analyzing a person's perception of wine is challenging due to the complex nature of flavor, which remains ill-understood, and the difficulty in obtaining consistent verbal descriptions of taste across individuals. Napping [Pagès, 2005] is the *de facto* method to analyze perceived taste in consumer surveys. Participants receive taste samples and are instructed to place them on a sheet of paper based on how similar they perceive their taste to be, with closer meaning more similar. Such experiments are usually conducted with 10-25 participants and less than 20 variants of a product [Giacalone et al., 2013, Pagès et al., 2010, Mayhew et al., 2016]. In this study, we scale this data collection process to 256 participants and 108 unique bottlings of red wine, resulting in over 400 napping papers collected and more than 5k annotated flavor distances. In contrast to previous works [Giacalone et al., 2013, Pagès et al., 2010, Mayhew et al., 2016] our objective is to incorporate taste as one of the modalities that contribute to the shared representations for improved grounding of machine learning models.

**Human kernel learning.** Annotating flavor with Napping [Pagès, 2005] does not provide image-flavor or text-flavor correspondences but rather relative flavor similarities between sampled products. According to [Miller, 2019] humans are better at describing abstract concepts such as taste with contrastive questions, such as *"does wine X taste more similar to wine Y or Z?"* For this reason, the machine learning community has used contrastive questions in multiple settings, e.g., for understanding how humans perceive light reflection from surfaces by presenting annotators with image triplets depicting the Stanford Bunny with varying material properties [Agarwal et al., 2007], to produce a genre embedding of musical artists [Van Der Maaten and Weinberger, 2012], and for discovering underlying narratives in online discussions [Christensen et al., 2022]. Most relevant to our work is SNaCK [Wilber et al., 2015], which presents annotators with image triplets depicting foods and asked which two of them taste more similar, to obtain flavor triplets. They proposed to combine this high-level human flavor understanding with low-level image statistics to learn food concepts, e.g., that even though guacamole and wasabi look similar, their taste is not. Having humans annotate image triplets of foods works well for coarse concepts, but does not encompass nuanced differences in taste. In this work, we focus on the much finer-grained taste difference found in wines. These nuances and the complex nature of wine tasting, which involves taste *and* smell, are not easily conveyed through text or images.

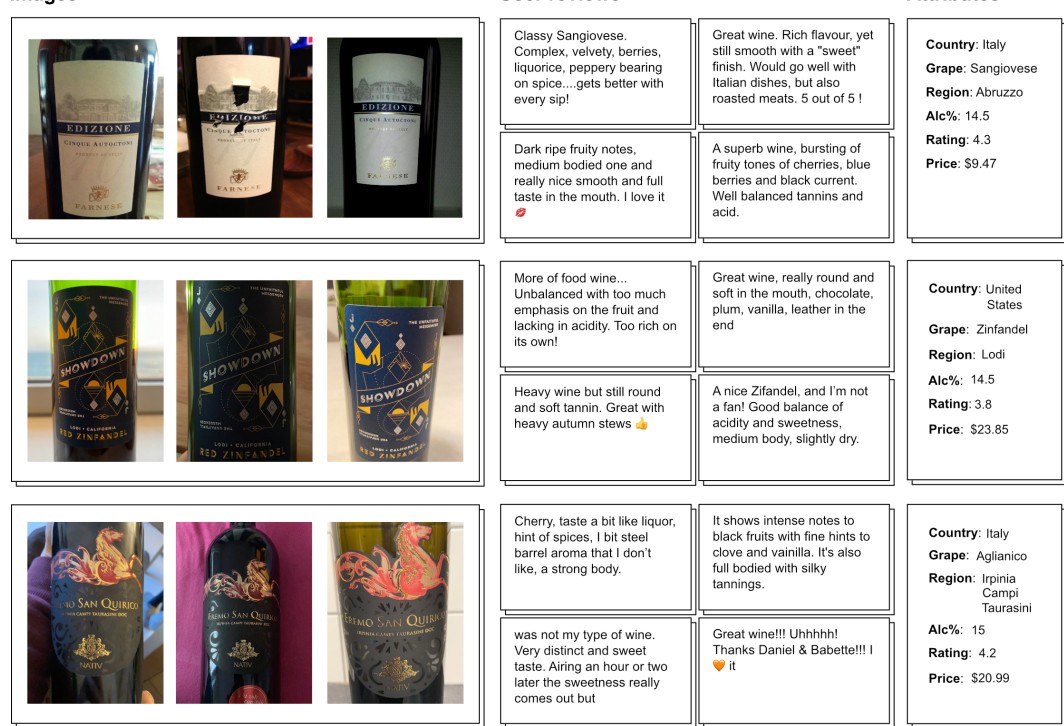

**Images**      **User reviews**      **Attributes**

Classy Sangiovese. Complex, velvety, berries, liquorice, peppery bearing on spice....gets better with every sip!

Great wine. Rich flavour, yet still smooth with a "sweet" finish. Would go well with Italian dishes, but also roasted meats. 5 out of 5 !

Dark ripe fruity notes, medium bodied one and really nice smooth and full taste in the mouth. I love it

A superb wine, bursting of fruity tones of cherries, blue berries and black current. Well balanced tannins and acid.

**Country**: Italy
**Grape**: Sangiovese
**Region**: Abruzzo
**Alc%**: 14.5
**Rating**: 4.3
**Price**: $9.47

More of food wine... Unbalanced with too much emphasis on the fruit and lacking in acidity. Too rich on its own!

Great wine, really round and soft in the mouth, chocolate, plum, vanilla, leather in the end

Heavy wine but still round and soft tannin. Great with heavy autumn stews

A nice Zifandel, and I'm not a fan! Good balance of acidity and sweetness, medium body, slightly dry.

**Country**: United States
**Grape**: Zinfandel
**Region**: Lodi
**Alc%**: 14.5
**Rating**: 3.8
**Price**: $23.85

Cherry, taste a bit like liquor, hint of spices, I bit steel barrel aroma that I don't like, a strong body.

It shows intense notes to black fruits with fine hints to clove and vanilla. It's also full bodied with silky tannings.

was not my type of wine. Very distinct and sweet taste. Airing an hour or two later the sweetness really comes out but

Great wine!!! Uhhhhh! Thanks Daniel & Babette!!! I ❤ it

**Country**: Italy
**Grape**: Aglianico
**Region**: Irpinia Campi Taurasini
**Alc%**: 15
**Rating**: 4.2
**Price**: $20.99

Figure 2: **Examples from WineSensed.** The dataset consists of images of wine labels, user-generated reviews, per-wine attributes (country, grape, region, alcohol percentage, rating, price), and flavor annotations. Here are examples of the images, reviews, and attributes.

**Flavor datasets.** The machine learning community has produced numerous food datasets for classifying which meal is in an image [Bossard et al., 2014, Min et al., 2020], retrieving a recipe given an image [Salvador et al., 2017, Li et al., 2022], or predicting the origin of wines [Dua and Graff, 2017]. While it is possible to extract coarse information about taste from such datasets [Wilber et al., 2015], they do not encompass higher resolution details of taste, such as the differences between a Cabernet Sauvignon and Pinot Noir.

Similarly, the food science community has developed many datasets for understanding and predicting food flavors, nutrient content, and chemistry. Flavornet [Arn and Acree, 1998], a dataset on human-perceived aroma compounds, explores partly how smells relate to perceived bitterness or fruitiness in a wine. However, its limitation is its lack of context linking these odors to specific wine varieties and its limited focus on flavor aspects. FoodDB [Harrington et al., 2019] offers comprehensive information on a wide variety of food, its nutrient contents, potential health effects, and macro and micro constituents. However, it lacks user-generated reviews and sensory data, which are crucial for understanding the subjective human perception of food and wine. The Wine Data Set [Dua and Graff, 2017] focuses on wines, but only contains wines originating from one region in Italy, limiting the dataset's ability to capture the broader diversity of flavor profiles of wines from various regions worldwide. Furthermore, Dua and Graff [2017] solely incorporate the chemical compounds present in each wine, without annotations of flavors and information associating specific wines with each chemical compound. In contrast to previous work, we present a multimodal dataset that contains a large corpus of images and reviews, as well as human-annotated flavor similarities.

## 3 The *WineSensed* dataset

We present WineSensed, a large, multimodal wine dataset that combines human flavor annotations, images, and reviews. In this section, we provide an overview of the curation process for each of these modalities.

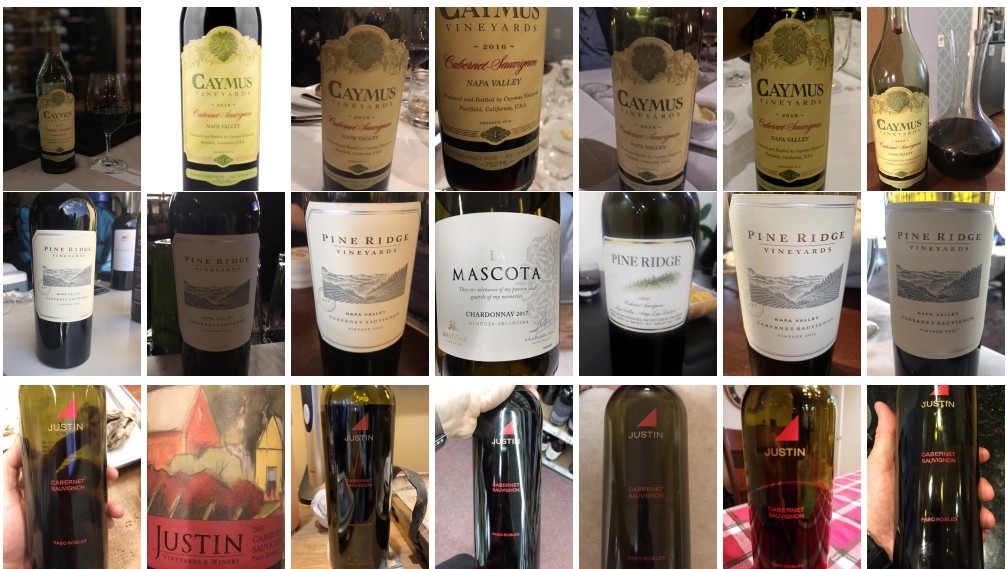

Figure 3: **Examples of images.** The viewpoint, lighting, and composition vary across images.

**Annotated flavors.** The flavor data consists of over 5k human-annotated pairwise similarities between 108 unique bottlings. Each annotated pair is annotated at least five times to reduce noise.

These annotations are collected through a series of wine-tasting events attended by a total of 256 non-expert wine drinkers. Most participants were between 21-25 years old, and more than half of them were from Denmark. Each participant volunteered their time, dedicating a maximum of two hours to complete the annotations. The experiment was conducted in accordance with the "De Videnskabsetiske Komiteer" (e. the Danish ethics committee for science) (see Appendix I).

We randomly selected 5 wines for the participants to taste. The participants did not have access to any information regarding the individual wines. The wine was poured into non-transparent shot glasses and the labels of the wines were covered during the entire experiment. The participants were instructed to put colored stickers (representing each of the five wines) on a sheet of paper based on their taste similarity, closer meaning more similar. The participants could repeat the process up to three times, ensuring they did not consume more than 225 ml of wine. The average participant repeated the experiment two times.

We automatically digitized the participants' annotations by taking a photo of each filled-out sheet. We used the Harris corner detector [Harris et al., 1988] to find the corners of the paper and a homographic projection to obtain an aligned top-down view of the paper. The images were mapped into HSV color space and a threshold filter applied to find the different colored stickers that the participant used to represent the wines. Having identified the location, we computed the Euclidean pixel-wise distance between all pairs of points, resulting in a distance matrix of wine similarities. A more detailed description of the collection and digitization of the napping papers can be found in D.

**User-reviews.** We curated 824k text reviews from the Vivino platform. The reviews were filtered to contain at least 10 characters to avoid non-informative reviews such as 'good' and 'bad.' Fig. 2 shows examples of user-reviews. The reviews are free text and can contain special tokens such as emojis. The reviews tend to describe price, pairing, and general terms of wine. Some also describe which flavors the reviewer tastes. These reviews are subjective and can vary based on personal factors and context, leading to inconsistent flavor profiles. Moreover, they only contain coarse flavor descriptions and focus more on aspects like preference, price, occasion, and so forth. Fig. 4 shows the distribution of word count per review, number of reviews per unique bottling, and the most common keywords.

**Images.** The dataset has 897k images of wine labels. Wine labels are known to play a major role in a consumer's decision to purchase a particular wine, so it is reasonable to believe that label design

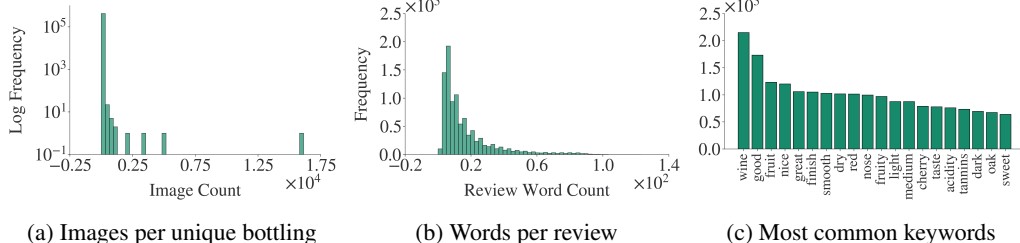

(a) Images per unique bottling      (b) Words per review      (c) Most common keywords

Figure 4: **Summary statistics of user reviews and images.** Most unique bottlings have less than 10 images. The average review length is 16 words. Common keywords in the reviews include 'fruit', 'dry', and 'smooth' revealing coarse semantic information about the flavor of the wines while other keywords such as 'good' and 'great' do not reveal flavor information.

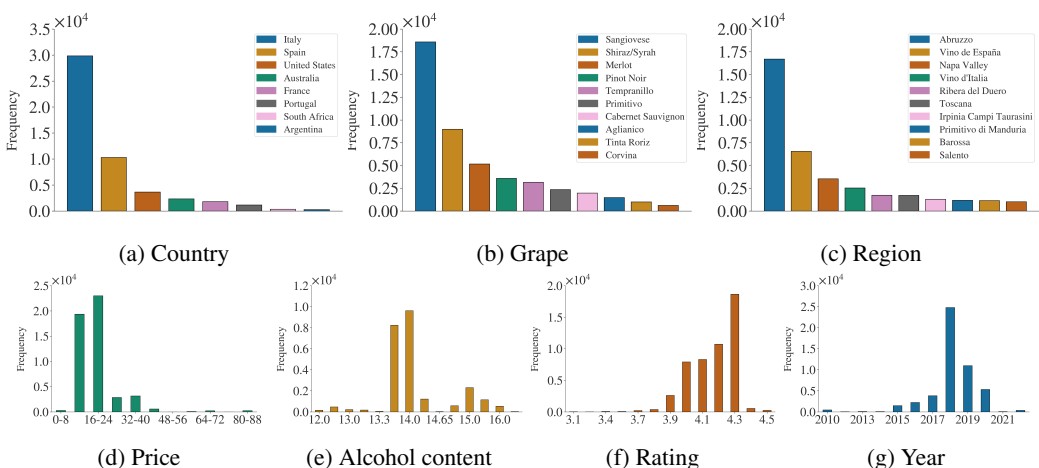

(a) Country      (b) Grape      (c) Region

(d) Price    (e) Alcohol content    (f) Rating    (g) Year

Figure 5: **Wine attributes.** WineSensed contains attributes about the geolocation of production (country, region) and the grape composition of each wine. Furthermore, the dataset includes information on the average price of the wine, alcohol percentage, average rating on the Vivino platform, and the year of production. The histograms show the distribution of these attributes.

carries information regarding the taste of the wine [Talbot, 2019]. Fig. 3 shows examples of images from the dataset. The images vary in their viewing angle, illumination, and image composition.

**Attributes.** Each wine is associated with the geographical location of the vineyard (both country and region), grape varietal composition, vintage, alcohol content, pricing, and average user rating. Fig. 5 shows the distribution of these attributes. Most wines originate from Italy, with Sangiovese being the most commonly used grape. The wines occupy the lower range of the price spectrum, with the most expensive ones priced at around 40 USD. The attributes are available for 5% of the dataset entries.

## 4 Flavor Embeddings from Annotated Similarity & Text-Image (FEAST)

The embeddings of recent large image and text networks contain structural and semantic information, however, they do not model the intricacies of human flavor. We propose FEAST, a method to align these embeddings to the human perception of flavor using a small set of human-annotated flavor similarities. FEAST takes text and/or images as input, as well as human-annotated flavor similarities. It outputs a unified embedding that aligns with human sensory perception. Fig. 6 provides an overview of the proposed method.

We first embed the text and/or images into a latent space with CLIP [Radford et al., 2021a]. We use CLIP because of its large training corpus and its image-text aligned latent space, however, highlights that other pretrained networks can be used. We use t-SNE [Van der Maaten and Hinton, 2008] to

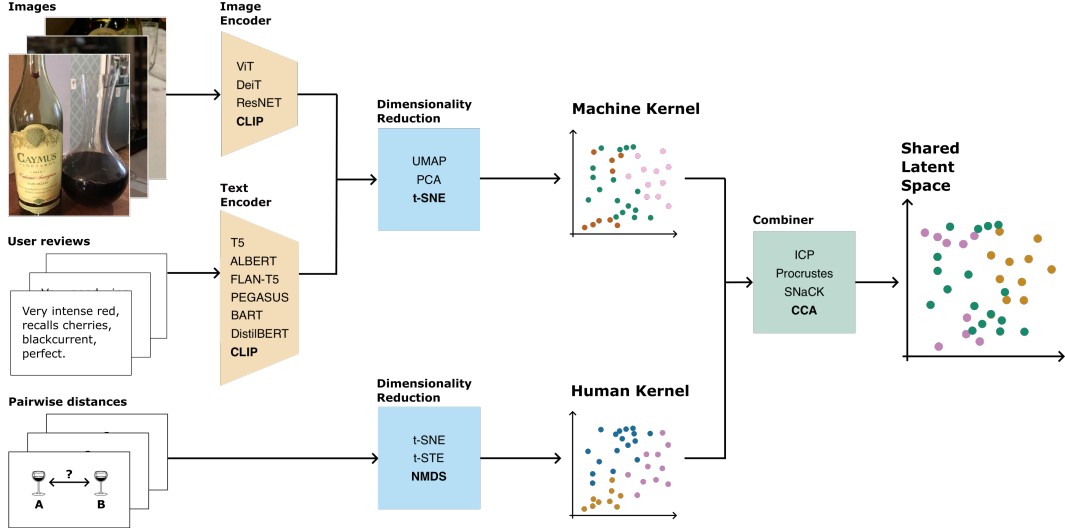

Figure 6: **Model overview.** FEAST takes text and/or images as input as well as human-annotated flavor similarities. The text and/or images are embedded into a latent representation with CLIP. We use NMDS to embed the flavor similarities. The two representations are aligned with CCA to produce a latent space that uses the structural information in CLIP embeddings and the intricacies of human annotations. The bolded methods in the orange, blue, and green boxes indicate choices for our best model, and their remaining combinations serve as an overview of the evaluated baselines.

reduce the dimensionality of the latent space to 2, which simplifies and constrain the later alignment with the pairwise flavor annotations.

The pairwise distances are embedded into a 2D representation using Non-metric multidimensional scaling (NMDS) with the SMACOF strategy [de Leeuw and Mair, 2009]. NMDS allows us to preserve the original flavor distances provided by humans in a shared space, where each unique bottling is represented with point location, rather than pairwise distances. MDS is commonly used in food science to analyze sensory annotations from Napping studies [Pineau et al., 2022, Varela and Ares, 2012, Nestrud and Lawless, 2010].

We then align these two 2D representations to get a joint representation that benefits from the structural and semantic information of the image and/or text representations, scales to unobserved unique bottlings, and is aligned with the human perception of flavor. We use Canonical Correlation Analysis (CCA) [Harold, 1936] to align the two representations. CCA identifies and connects common patterns between these representation spaces, ensuring that the final representation is consistent across all input modalities.

## 5   Experiments

We conduct two experiments on the WineSensed dataset. First, we explore how well recent large pretrained language and image models explain wine attributes that correlate with the flavor of a wine. Second, we explore multimodal models' capabilities to represent more intricate flavors.

**Experimental setup.** We explore several configurations of human kernels, machine kernels, and "combiners" that align the two representations. Fig. 6 provides an overview of our baselines. The **human kernel** is formed with t-STE [Van Der Maaten and Weinberger, 2012], a low dimensional graph representation reduced with t-SNE or NMDS, where the notable difference is that t-STE discards the flavor distances, and solely optimizes for triplet orderings. The **machine kernel** consists of two steps: (1) we use a pretrained model to embed text and/or images into a low dimensional space, (2) which is then compressed into a two-dimensional space. For (1), we explore DistilBert

Table 1: **Ablation of machine kernels.** Accuracy of machine kernels across image and text modalities. Image models perform worse than text models. ALBERT, BART and CLIP perform the best, all models perform better than random using at least one classification method.

| Machine kernel | Modality | Acc ↑ SVM | NN |
|---|---|---|---|
| Random | | 0.11 | 0.11 |
| ViT | Image | 0.09 | 0.13 |
| DeiT | Image | 0.14 | 0.15 |
| ResNET | Image | 0.15 | 0.16 |
| CLIP | Image | 0.11 | 0.15 |
| T5 | Text | 0.15 | 0.16 |
| **ALBERT** | **Text** | 0.15 | **0.18** |
| **BART** | **Text** | **0.16** | 0.15 |
| DistilBERT | Text | 0.15 | 0.17 |
| **CLIP** | **Text** | **0.16** | **0.18** |
| FLAN-T5 | Text | 0.15 | 0.17 |
| PEGASUS | Text | 0.13 | 0.13 |
| BART | Text | 0.11 | 0.15 |

Table 2: **Ablation of Modalities.** Accuracy of single and combined modalities. Using multiple modalities improves performance. We find that combining image, text, and flavor yields much better accuracy than modeling each modality separately.

| Modality | Acc ↑ SVM | NN |
|---|---|---|
| Flavor | 0.16 | 0.11 |
| Image | 0.11 | 0.15 |
| Text | 0.16 | 0.18 |
| Text+Flavor | 0.23 | 0.18 |
| Image+Text | 0.22 | 0.25 |
| Image+Flavor | 0.23 | 0.18 |
| **Image+Text+Flavor** | **0.28** | **0.26** |

Table 3: **Ablation of human kernels, reducers, and combiners.**

| Reducer Human Kernel | Acc ↑ SVM | NN |
|---|---|---|
| Random | 0.11 | 0.11 |
| t-STE | 0.13 | 0.10 |
| t-SNE | 0.15 | 0.13 |
| **NMDS** | **0.16** | **0.13** |

| Reducer Machine Kernel | Acc ↑ SVM | NN |
|---|---|---|
| UMAP | 0.15 | 0.18 |
| PCA | 0.20 | 0.21 |
| **t-SNE** | **0.22** | **0.25** |

| Combiner | Acc ↑ SVM | NN |
|---|---|---|
| ICP | 0.21 | 0.24 |
| Procrustes | 0.19 | 0.23 |
| SNaCK | 0.23 | 0.24 |
| **CCA** | **0.28** | **0.26** |

[Sanh et al., 2019], T5 [Raffel et al., 2020], ALBERT [Lan et al., 2019], BART [Lewis et al., 2019], PEGASUS [Zhang et al., 2020], FLAN-T5 [Chung et al., 2022] and CLIP for embedding text and ViT [Dosovitskiy et al., 2020], ResNet [He et al., 2016], DeiT [Touvron et al., 2021], and CLIP for embedding images. For (2), we explore t-SNE, UMAP [McInnes et al., 2018], and PCA [Pearson, 1901]. For the **combiners**, we experiment with CCA, Iterative Closest Point (ICP) [Chen and Medioni, 1992], Procrustes [Gower, 1975] and SNaCK. For a more detailed description of the implementation and software packages used, please refer to E the Appendix.

## 5.1 Coarse flavor predictions

We first explore how well pretrained language and vision models explain wine attributes that correlate with flavor. We then investigate if using FEAST to align the machine and human kernels improves the representation.

**Implementation details.** We use a balanced SVM classifier with an RBF kernel as well as a Multi-layer Perceptron [] neural network to predict wine attributes of the flavor embeddings. We predict price, alcohol percentage, rating, region, country, and grape variety as these attributes are known to correlate with the perceived wine flavor. We mitigate imbalanced class distributions with class weight balancing and oversampling of the minority classes. We report the accuracy averaged over the seven attributes computed through 5-fold cross-validation. The accuracy measures how coherent the embeddings are with the flavor attributes. A more detailed description of the implementation can be found in J.2.

**Results.** Tables 1 to 3 ablates our proposed method and summarizes our main conclusions. Please see Appendix J.2 for per attribute classification accuracy for all combinations of machine kernels, human kernels, modalities, reduces, and combiners.

Table 1 shows that most pretrained image and text models yield slightly higher performance than the random baseline. The text encoders are slightly better than the image encoders. BART and CLIP perform the best. All encoders in the table use t-SNE to reduce the embedding to 2D. Table 3 (middle) shows t-SNE yields better accuracy than UMAP and PCA when using a CLIP encoder.

Table 3 (top) shows that NMDS performs better than t-STE. NMDS uses the relative distances between annotations, whereas t-STE discretizes the annotations and considers only the ordering within each triplet. The results suggest that the pairwise distances are useful to model the flavor space.

Table 4: **Fine-grained flavor predictions.** Triplet Agreement Ratio (TAR) between text, image, and multi-modal encoders and human annotated flavor similarities. A higher TAR indicates that the model's representation space is more aligned with humans' perception of flavor.

| Machine Kernel | Human Kernel | Combiner | Modality | TAR ↑ |
|---|---|---|---|---|
| Random | | | | 0.5 |
| CLIP + t-SNE | | | Text | 0.82 |
| CLIP + t-SNE | | | Image | 0.82 |
| CLIP + t-SNE | | | Image + Text | 0.81 |
| CLIP + t-SNE | | | Image + Flavor | 0.89 |
| CLIP + t-SNE | | | Text + Flavor | 0.88 |
| CLIP + t-SNE | NMDS | CCA | Image + Text + Flavor | **0.91** |

Table 3 (bottom) shows that using CCA to align the two representations yields higher accuracy than SNaCK or ICP.

Table 2 shows that including flavor as a modality increases the accuracy, *e.g.* using flavor to align the image or text embeddings lead to higher accuracy. Using CLIP followed by t-SNE, NMDS, and CCA to combine language, vision, and flavor into a single representation leads to the best configurations, illustrating that the human annotations are useful for learning a flavor representation. Maybe most surprisingly, we show that each modality by itself is on par with the random baseline, but their combination produces a latent space that much better describes the flavor attributes.

## 5.2 Fine-grained flavor predictions

We now proceed to evaluate more intricate flavor predictions by using human-annotated flavor similarities as ground truth.

**Implementation details.** To evaluate our representation, we measure the Triplet Agreement Ratio (TAR) [van der Maaten and Weinberger, 2012] between our predicted flavor embeddings and the human-annotated flavors. TAR measures the agreement between a triplet derived from the latent space and the ground truth triplets from the flavor annotations. Higher TAR means that the ordering of distances in the latent space corresponds to the human perception of flavor. This measure indicates how aligned the two representations are, and provides a higher granularity of flavor prediction than flavor attributes. A more detailed description of the implementation can be found in F.

**Results.** Table 4 ablates FEAST and shows that for the higher granularity predictions both the pretrained text and image encoders improve upon the random baseline. We show that including the human kernel with NMDS further improves the TAR scores. This highlights the usefulness of the flavor distances recorded by the human annotators. In Appendix F, we show results from all configurations of human kernels, machine kernels, reducers, and combiners. We find that NMDS consistently yields better performance than t-STE, and that combining human and machine kernels improves the TAR scores across multiple model configurations.

## 6 Discussion & Conclusion

In this paper, we introduce WineSensed, an extensive multimodal dataset curated for flavor modeling. The dataset comprises over 897k images and 824k reviews, and has over 5k human-annotated pairwise flavor similarities, obtained via a sensory study involving 256 participants. We propose a simple algorithm, FEAST, to align semantic information from machine kernels with flavor similarities from human annotators in a shared flavor representation. We find that combining these modalities improves both coarse and fine-grained flavor predictions.

WineSensed further strengthens the collaboration between the food science and machine learning communities, introduces flavor as a modality in multimodal models, and serves as an entry point for the development of machine learning models for flavor analysis and potentially deepening our

comprehension of wine flavors. The dataset and the proposed procedures open many interesting possibilities, such as using flavor to ground foundation models or extending the dataset with other modalities, such as chemical composition, or other food categories.

**Constraints and considerations.** The dataset serves as a novel first step to including human-annotated flavor in the array of modalities in multimodal models. Its current scope is constrained to a selected group of red wines, predominantly Italian ones. While this enables a more nuanced understanding of flavors within Italian wines, it may not represent the broader spectrum of red wines globally. Furthermore, the dataset's emphasis on wines prevalent in Western cultures highlights a geo-cultural bias. Expanding the dataset to encompass more diverse drink types from different cultures could provide a more comprehensive understanding of global flavor perception. Lastly, the Napping methodology is not immune to the influences of participants' backgrounds and experiences. Individual perceptions, shaped by personal histories, can introduce nuances in the data. Though leveraging non-expert wine drinkers for flavor annotations introduces subjectivity, this approach, inspired by common sensory study practices, broadens taste perspectives, enhances study accessibility, and offers commercial value, with multiple annotations per entry mitigating individual biases. Exploring a broader range of foods and beverages remains a valuable direction for future work.

**Acknowlegements.** This work was supported by the Pioneer Centre for AI, DNRF grant number P1, and by research grant (42062) from VILLUM FONDEN. This project received funding from the European Research Council (ERC) under the European Union's Horizon 2020 research and innovation programme (grant agreement 757360), as well as the Danish Data Science Academy (DDSA).

