## A Project webpage

We provide a project webpage for the dataset that can be found here: `https://thoranna.github.io/learning_to_taste/`, which contains a link to the dataset and the code to reproduce our experiments. Additionally, we provide more examples from our dataset and images from the data collection.

## B The WineSensed file structure

Our dataset is currently available here: `https://data.dtu.dk/articles/dataset/WineSensed_Learning_to_Taste_A_Multimodal_Wine_Dataset/23376560`. The dataset will be maintained on this site, which is hosted on a server run by the Technical University of Denmark.

WineSensed contains a `metadata.zip` file consisting of the files `participants.csv`, which contains information connecting participants to annotations in the experiment, `images_reviews_attributes.csv`, which contains reviews, links to images, and wine attributes, and `napping.csv`, which contains the coordinates of each wine on the napping paper, alongside information connecting each coordinate pair to the wines being annotated and the participant that annotated them. The `chunk_<chunk num>.zip` folders contain the images of the wines in the dataset in `.jpg` format.

`napping.csv` contains the following fields:

- `session_round_name`: session number during the `event_name`, at most three sessions per event (maps to `experiment_round` in `participants.csv`)
- `event_name`: name of the data collection event (maps to the same attribute in `participants.csv`)
- `experiment_no`: the serial number of the napping paper in the `session_round_name` in which it was collected (maps to `experiment_no` in `participants.csv`)
- `experiment_id`: id of the wine annotated
- `coor1`: x-axis coordinate on the napping paper
- `coor2`: y-axis coordinate on the napping paper
- `color`: color of the sticker used

`participants.csv` contains the following fields:

- `session_round_name`: session number during the `event_name`, at most three sessions per event (maps to `experiment_round` in `napping.csv`)
- `event_name`: name of data-collection event (maps to `event_name` in `napping.csv`)
- `experiment_no`: the serial number of the napping paper in the `session_round_name` in which it was collected (maps to `experiment_no` in `napping.csv`)
- `round_id`: round number (from 1-3)
- `participant_id`: id the participant was given in the experiment

`images_reviews_attributes.csv` contains the following fields:

- `vintage_id`: vintage id of the wine
- `image`: image link (each `<image name>.jpg` in `chunk_<chunk num>.zip` can be mapped to a corresponding image link in this column by removing the `/p` prefix from the link).
- `review`: user review of the wine

- `experiment_id`: id the wine got during data collection (each `experiment_id` can be mapped to the same column in `napping.csv`)

- `year`: year the wine was produced

- `winery_id`: id of the winery that produced the wine

- `wine`: name of the wine

- `alcohol`: the wine's alcohol percentage

- `country`: the country where the wine was produced

- `region`: the region where the wine was produced

- `price`: price of the wine in USD (collected 05/2023)

- `rating`: average rating of the wine (collected 05/2023)

- `grape`: the wine's grape composition, represented as a comma-separated list ordered in descending sequence of the percentage contribution of each grape variety to the overall blend.

## C   Implementation details for flavor space generation

**Preprocessing.**   For the image data, we resized images to a 256x256 pixel format, applied a central crop to bring the images down to 224x224 pixels. Subsequently, we converted them into a tensor format, followed by normalization using mean and standard deviation values for each color channel (RGB).

For the user reviews, we first converted the text to lowercase to maintain consistency. Then, we removed punctuation marks to minimize noise. We further eliminated stopwords using the nltk library's English stopword list since these words usually do not contribute significantly to the overall meaning of the reviews. After these preprocessing steps, the data was tokenized and reassembled into a clean text string.

The preprocessing of human-annotated data varied based on its intended use, either as a distance matrix or triplets. In the former case, we calculated the Euclidean distances between each data point and arranged these distances into an $N \times N$ matrix, where $N$ is the total number of annotated wines. The matrix element $m[i][j]$ had a value of $0$ if there were no annotated distances between wines $i$ and $j$. For the latter scenario, we constructed a list of triplets derived from the computed Euclidean distances. We generated triplets $(i, j, k)$ based on the Euclidean distances, such that $i$ is closer to $j$ than to $k$; i.e. $\|i - j\|_2 < \|i - k\|_2$.

**Dimensionality reduction.**   In our experiments, we used several dimensionality reduction methods such as NMDS, t-STE, t-SNE, PCA, and UMAP. For these methods, we prepared two embedding pipelines, one to reduce the dimensionality of machine kernel, and another to reduce the dimensionality of the human kernel.

For the human kernel, NMDS and t-STE were used. The NMDS method was optimized through a series of hyperparameter tunings, including number of initial positions (`n_inits`), maximum number of iterations (`max_iters`), and tolerance to stress convergence (`eps_values`). These hyperparameters were evaluated using a range of values with the number of initial positions set to 5, 7, 10, the maximum number of iterations set to 300, 400, 500, 600, and the tolerance for stress convergence set to 1e-3, 1e-4, 1e-5.

The optimal hyperparameters for NMDS were selected by applying 5-fold cross validation (`cross_val_score`) using a K-nearest neighbors classifier model (`KNeighborsClassifier`) and oversampling to handle class imbalances in the data. In NMDS, The parameter `metric` was set to `False` to handle dissimilarities missing values represented by zeroes, and `dissimilarity` to precomputed as the input data was a distance matrix. Classification improvements during grid-search were not significant.

For the machine kernel pipeline, t-SNE, PCA, and UMAP, were used with a set seed to ensure the results' reproducibility. These methods were called using their default hyperparameters in the respective libraries (see External packages).

**Pre-trained models.** The machine kernel embeddings were obtained using a collection of pre-trained text, image, and combined image-text models. All models were obtained from the HuggingFace [hug] library. The chosen models for the text were T5 (60.5M params), ALBERT (11.8M params), BART (139M params), DistilBERT (67M params), and CLIP text model. For images, we chose ViT, DeiT, ResNET-50 and the CLIP image encoder. Lastly, we used CLIP for the combined image-text model. All embeddings were obtained from the models' last hidden state.

**Combiners.** We leveraged three methods to combine the human kernel and the machine kernel: CCA, ICP, and SNaCK. These three methods were employed using their default hyperparameters in their respective libraries (see External packages). In the case of CCA and ICP, we found common experiment identifiers across the two datasets and used them to align corresponding data points from the two datasets. Once the matrices were aligned, we subsequently applied CCA and ICP, respectivelt, and generated the combined embeddings thus.

SNaCK follows a slightly different process as it uses triplets from the human kernel and an embedding matrix from the machine kernel. We passed the triplet list (human kernel) and scaled embeddings (machine kernel) into SNaCK, which output the combined embedding.

**External packages.** We used several external packages: `scikit-learn` (v1.2.2) [sci], for dimensionality reduction, hyperparameter optimization, classification and human-and machine kernel combination; `umap-learn` (v0.5.3) [uma], for dimensionality reduction of the machine kernel; `imblearn` (v0.10.1) [imb], to address the problem of imbalanced datasets; `snack` sna, an implementation of SNaCK for human-and machine kernel combination; `icp` [icp], implementing the Iterative Closest Point algorithm for human-and machine kernel combination; and `tste` [tst], an implementation of the t-Distributed Stochastic Triplet Embedding algorithm for the dimensionality reduction of human-kernel triplets. Additionally, our project employed these Python packages: `torchmetrics` (v0.11.4) tor, `ftfy` (v6.1.1) [ftf], `open-clip-torch` (v2.19.0) [ope], `transformers` (v4.28.1) tra, `pandas` (v2.0.1) [pan], `nltk` (v3.8.1) [nlt], `psutil` (v5.9.5) [psu], `urllib3` (v1.26.15) [url], `matplotlib` (v3.5.1) mat, `seaborn` (v0.11.2) [sea], and `h5py` (v3.8.0) [h5p].

# D Details for fine-grained flavor predictions

**Implementation details.** The combination of dimensionality reduction methods, pre-trained models, and combiners described in D were used to generate multiple flavor spaces (using images, text and flavor). Additionally, to compare TAR across modalities, embeddings were produced for all combinations of modalities (text, image and flavor) using the relevant methods from D.

The human kernel was split into a training and a testing set. We made sure that for any given triplet $(i, j, k)$ in the testing set, none of the wines $i$, $j$ or $k$ were present in the training set. The training set was processed and combined with the machine kernel using the reduction methods and combiners from D. The triplet agreement ratio was calculated using the level of agreement between the testing set and the triplets in the embeddings, by dividing agreements with disagreements. The triplet agreement ratio's random baseline was set at 0.5, because when comparing triplets, either (i, j, k) or (j, i, k) could be chosen, which makes the ratio 0.5/1.0, similar to a random guess.

**Results.** All results produced in this experiment can be found in tables 5, 6 and 7.

# E Details for coarse-grained flavor predictions.

# F Implementation Details

We utilize a SVM classifier with parameter `class_weight` set to balanced and and K-fold cross-validation with `n_splits` set to 5 and `shuffle` set to True using the classifier `SVC` and the method

Table 5: **Fine-grained flavor predictions: Text encoders.** Triplet Agreement Ratio (TAR) between text encoders and human annotated flavor similarities.

| Machine Kernel | Human Kernel | Combiner | Modality | TAR ↑ |
|---|---|---|---|---|
| DistilBeRT + UMAP | | | Text only | 0.81 |
| DistilBeRT + t-SNE | | | Text only | 0.81 |
| DistilBeRT + UMAP | MDS | CCA | Text + flavor | **0.91** |
| DistilBeRT + t-SNE | MDS | ICP | Text + flavor | 0.90 |
| DistilBeRT + t-SNE | MDS | CCA | Text + flavor | 0.90 |
| DistilBeRT + UMAP | t-STE | CCA | Text + flavor | 0.76 |
| DistilBeRT + t-SNE | t-STE | ICP | Text + flavor | 0.78 |
| DistilBeRT + t-SNE | t-STE | SNaCK | Text + flavor | 0.75 |
| T5 + UMAP | | | Text only | 0.82 |
| T5 + t-SNE | | | Text only | 0.82 |
| T5 + UMAP | MDS | CCA | Text + flavor | 0.89 |
| T5 + t-SNE | MDS | ICP | Text + flavor | **0.90** |
| T5 + t-SNE | MDS | CCA | Text + flavor | **0.90** |
| T5 + UMAP | t-STE | CCA | Text + flavor | 0.83 |
| T5 + t-SNE | t-STE | ICP | Text + flavor | 0.78 |
| T5 + t-SNE | t-STE | SNaCK | Text + flavor | 0.84 |
| ALBERT + UMAP | | | Text only | 0.80 |
| ALBERT + t-SNE | | | Text only | 0.81 |
| ALBERT + UMAP | MDS | CCA | Text + flavor | 0.89 |
| ALBERT + t-SNE | MDS | ICP | Text + flavor | **0.90** |
| ALBERT + t-SNE | MDS | CCA | Text + flavor | **0.90** |
| ALBERT + UMAP | t-STE | CCA | Text + flavor | 0.74 |
| ALBERT + t-SNE | t-STE | ICP | Text + flavor | 0.78 |
| ALBERT + t-SNE | t-STE | SNaCK | Text + flavor | 0.78 |
| BART + UMAP | | | Text only | 0.81 |
| BART + t-SNE | | | Text only | 0.82 |
| BART + UMAP | MDS | CCA | Text + flavor | 0.89 |
| BART + t-SNE | MDS | ICP | Text + flavor | **0.90** |
| BART + t-SNE | MDS | CCA | Text + flavor | 0.89 |
| BART + UMAP | t-STE | CCA | Text + flavor | 0.78 |
| BART + t-SNE | t-STE | ICP | Text + flavor | 0.79 |
| BART + t-SNE | t-STE | SNaCK | Text + flavor | 0.72 |

KFold from the Scikit-Learn library [sci]. Additionally we utilize RandomOverSampler from the imblearn library with sampling_strategy set to 'not majority'. When dealing with non-numerical attributes, a LabelEncoder (using the default values) from Scikit-Learn [sci] was used to create numerical features. The random baseline value was calculated by dividing 1 by the number of classes to predict.

**Results.** All results produced in this experiment can be found in tables 8, 9, 10, 11, 12, 13, 14, 15 and 16.

Table 6: **Fine-grained flavor predictions: Image encoders.** Triplet Agreement Ratio (TAR) between image encoders and human annotated flavor similarities.

| Machine Kernel | Human Kernel | Combiner | Modality | TAR ↑ |
|---|---|---|---|---|
| ViT + UMAP | | | Image only | 0.83 |
| ViT + t-SNE | | | Image only | 0.82 |
| ViT + UMAP | MDS | CCA | Image + flavor | **0.90** |
| ViT + t-SNE | MDS | ICP | Image + flavor | **0.90** |
| ViT + t-SNE | MDS | CCA | Image + flavor | **0.90** |
| ViT + UMAP | t-STE | CCA | Image + flavor | 0.82 |
| ViT + t-SNE | t-STE | ICP | Image + flavor | 0.78 |
| ViT + t-SNE | t-STE | SNaCK | Image + flavor | 0.75 |
| ResNET + UMAP | | | Image only | 0.82 |
| ResNET + t-SNE | | | Image only | 0.82 |
| ResNET + UMAP | MDS | CCA | Image + flavor | 0.89 |
| ResNET + t-SNE | MDS | ICP | Image + flavor | **0.90** |
| ResNET + t-SNE | MDS | CCA | Image + flavor | 0.88 |
| ResNET + UMAP | t-STE | CCA | Image + flavor | 0.79 |
| ResNET + t-SNE | t-STE | ICP | Image + flavor | 0.78 |
| ResNET + t-SNE | t-STE | SNaCK | Image + flavor | 0.76 |
| DeiT + UMAP | | | Image only | 0.82 |
| DeiT + t-SNE | | | Image only | 0.83 |
| DeiT + UMAP | MDS | CCA | Image + flavor | 0.91 |
| DeiT + t-SNE | MDS | ICP | Image + flavor | 0.90 |
| DeiT + t-SNE | MDS | CCA | Image + flavor | **0.92** |
| DeiT + UMAP | t-STE | CCA | Image + flavor | 0.82 |
| DeiT + t-SNE | t-STE | ICP | Image + flavor | 0.78 |
| DeiT + t-SNE | t-STE | SNaCK | Image + flavor | 0.86 |
| CLIP + UMAP | | | Image only | 0.82 |
| CLIP + t-SNE | | | Image only | 0.82 |
| CLIP + UMAP | MDS | CCA | Image + flavor | 0.89 |
| CLIP + t-SNE | MDS | ICP | Image + flavor | **0.90** |
| CLIP + t-SNE | MDS | CCA | Image + flavor | **0.90** |
| CLIP + UMAP | t-STE | CCA | Image + flavor | 0.81 |
| CLIP + t-SNE | t-STE | ICP | Image + flavor | 0.78 |
| CLIP + t-SNE | t-STE | SNaCK | Image + flavor | 0.81 |

Table 7: **Fine-grained flavor predictions: Text-Image encoder.** Triplet Agreement Ratio (TAR) between CLIP and human annotated flavor similarities.

| Machine Kernel | Human Kernel | Combiner | TAR Machine Kernel ↑ | TAR ↑ |
|---|---|---|---|---|
| CLIP + UMAP | | | Image + text | 0.82 |
| CLIP + t-SNE | | | Image + text | 0.81 |
| CLIP + UMAP | MDS | CCA | Image + text + flavor | **0.91** |
| CLIP + t-SNE | MDS | ICP | Image + text + flavor | 0.90 |
| CLIP + t-SNE | MDS | CCA | Image + text + flavor | **0.91** |
| CLIP + UMAP | t-STE | CCA | Image + text + flavor | 0.84 |
| CLIP + t-SNE | t-STE | ICP | Image + text + flavor | 0.78 |
| CLIP + t-SNE | t-STE | SNaCK | Image + text + flavor | 0.79 |

Table 8: **DistilBeRT:** Classification results.

| Machine Kernel | Human Kernel | Combiner | Class | Modality | Pred | ACC ↑ |
|---|---|---|---|---|---|---|
| Random | | | Country | | | 0.13 |
| DistilBeRT + UMAP | | | Country | Text only | SVM | 0.07 |
| DistilBeRT + t-SNE | | | Country | Text only | SVM | 0.19 |
| DistilBeRT + UMAP | MDS | CCA | Country | Text + flavor | SVM | **0.27** |
| DistilBeRT + t-SNE | MDS | ICP | Country | Text + flavor | SVM | 0.20 |
| DistilBeRT + t-SNE | MDS | CCA | Country | Text + flavor | SVM | 0.22 |
| DistilBeRT + UMAP | t-STE | CCA | Country | Text + flavor | SVM | 0.24 |
| DistilBeRT + t-SNE | t-STE | ICP | Country | Text + flavor | SVM | 0.20 |
| DistilBeRT + t-SNE | t-STE | SNaCK | Country | Text + flavor | SVM | 0.15 |
| Random | | | Region | | | 0.02 |
| DistilBeRT + UMAP | | | Region | Text only | SVM | 0.01 |
| DistilBeRT + t-SNE | | | Region | Text only | SVM | 0.02 |
| DistilBeRT + UMAP | MDS | CCA | Region | Text + flavor | SVM | 0.02 |
| DistilBeRT + t-SNE | MDS | ICP | Region | Text + flavor | SVM | 0.01 |
| DistilBeRT + t-SNE | MDS | CCA | Region | Text + flavor | SVM | **0.04** |
| DistilBeRT + UMAP | t-STE | CCA | Region | Text + flavor | SVM | 0.02 |
| DistilBeRT + t-SNE | t-STE | ICP | Region | Text + flavor | SVM | 0.01 |
| DistilBeRT + t-SNE | t-STE | SNaCK | Region | Text + flavor | SVM | 0.00 |
| Random | | | Grape | | | 0.03 |
| DistilBeRT + UMAP | | | Grape | Text only | SVM | 0.01 |
| DistilBeRT + t-SNE | | | Grape | Text only | SVM | 0.05 |
| DistilBeRT + UMAP | MDS | CCA | Grape | Text + flavor | SVM | **0.08** |
| DistilBeRT + t-SNE | MDS | ICP | Grape | Text + flavor | SVM | 0.05 |
| DistilBeRT + t-SNE | MDS | CCA | Grape | Text + flavor | SVM | 0.03 |
| DistilBeRT + UMAP | t-STE | CCA | Grape | Text + flavor | SVM | 0.07 |
| DistilBeRT + t-SNE | t-STE | ICP | Grape | Text + flavor | SVM | 0.04 |
| DistilBeRT + t-SNE | t-STE | SNaCK | Grape | Text + flavor | SVM | 0.04 |
| Random | | | Alc % | | | 0.17 |
| DistilBeRT + UMAP | | | Alc % | Text only | SVM | 0.12 |
| DistilBeRT + t-SNE | | | Alc % | Text only | SVM | 0.27 |
| DistilBeRT + UMAP | MDS | CCA | Alc % | Text + flavor | SVM | 0.45 |
| DistilBeRT + t-SNE | MDS | ICP | Alc % | Text + flavor | SVM | 0.34 |
| DistilBeRT + t-SNE | MDS | CCA | Alc % | Text + flavor | SVM | **0.49** |
| DistilBeRT + UMAP | t-STE | CCA | Alc % | Text + flavor | SVM | 0.48 |
| DistilBeRT + t-SNE | t-STE | ICP | Alc % | Text + flavor | SVM | 0.34 |
| DistilBeRT + t-SNE | t-STE | SNaCK | Alc % | Text + flavor | SVM | 0.10 |
| Random | | | Price | | | 0.10 |
| DistilBeRT + UMAP | | | Price | Text only | SVM | 0.11 |
| DistilBeRT + t-SNE | | | Price | Text only | SVM | 0.20 |
| DistilBeRT + UMAP | MDS | CCA | Price | Text + flavor | SVM | 0.21 |
| DistilBeRT + t-SNE | MDS | ICP | Price | Text + flavor | SVM | 0.14 |
| DistilBeRT + t-SNE | MDS | CCA | Price | Text + flavor | SVM | 0.18 |
| DistilBeRT + UMAP | t-STE | CCA | Price | Text + flavor | SVM | **0.30** |
| DistilBeRT + t-SNE | t-STE | ICP | Price | Text + flavor | SVM | 0.14 |
| DistilBeRT + t-SNE | t-STE | SNaCK | Price | Text + flavor | SVM | 0.11 |
| Random | | | Rating | | | 0.25 |
| DistilBeRT + UMAP | | | Rating | Text only | SVM | 0.25 |
| DistilBeRT + t-SNE | | | Rating | Text only | SVM | 0.27 |
| DistilBeRT + UMAP | MDS | CCA | Rating | Text + flavor | SVM | 0.45 |
| DistilBeRT + t-SNE | MDS | ICP | Rating | Text + flavor | SVM | 0.33 |
| DistilBeRT + t-SNE | MDS | CCA | Rating | Text + flavor | SVM | 0.48 |
| DistilBeRT + UMAP | t-STE | CCA | Rating | Text + flavor | SVM | **0.59** |
| DistilBeRT + t-SNE | t-STE | ICP | Rating | Text + flavor | SVM | 0.33 |
| DistilBeRT + t-SNE | t-STE | SNaCK | Rating | Text + flavor | SVM | 0.19 |
| Random | | | Year | | | 0.08 |
| DistilBeRT + UMAP | | | Year | Text only | SVM | 0.04 |
| DistilBeRT + t-SNE | | | Year | Text only | SVM | 0.08 |
| DistilBeRT + UMAP | MDS | CCA | Year | Text + flavor | SVM | **0.17** |
| DistilBeRT + t-SNE | MDS | ICP | Year | Text + flavor | SVM | 0.10 |
| DistilBeRT + t-SNE | MDS | CCA | Year | Text + flavor | SVM | 0.13 |
| DistilBeRT + UMAP | t-STE | CCA | Year | Text + flavor | SVM | 0.16 |
| DistilBeRT + t-SNE | t-STE | ICP | Year | Text + flavor | SVM | 0.10 |
| DistilBeRT + t-SNE | t-STE | SNaCK | Year | Text + flavor | SVM | 0.09 |

Table 9: **T5:** Classification results.

| Machine Kernel | Human Kernel | Combiner | Class | Modality | Pred | ACC ↑ |
|---|---|---|---|---|---|---|
| Random | | | Country | | | 0.13 |
| T5 + UMAP | | | Country | Text only | SVM | 0.05 |
| T5 + t-SNE | | | Country | Text only | SVM | 0.11 |
| T5 + UMAP | MDS | CCA | Country | Text + flavor | SVM | 0.11 |
| T5 + t-SNE | MDS | ICP | Country | Text + flavor | SVM | 0.08 |
| T5 + t-SNE | MDS | CCA | Country | Text + flavor | SVM | 0.13 |
| T5 + UMAP | t-STE | CCA | Country | Text + flavor | SVM | **0.18** |
| T5 + t-SNE | t-STE | ICP | Country | Text + flavor | SVM | 0.08 |
| T5 + t-SNE | t-STE | SNaCK | Country | Text + flavor | SVM | 0.08 |
| Random | | | Region | | | 0.02 |
| T5 + UMAP | | | Region | Text only | SVM | 0.01 |
| T5 + t-SNE | | | Region | Text only | SVM | 0.01 |
| T5 + UMAP | MDS | CCA | Region | Text + flavor | SVM | 0.01 |
| T5 + t-SNE | MDS | ICP | Region | Text + flavor | SVM | 0.00 |
| T5 + t-SNE | MDS | CCA | Region | Text + flavor | SVM | 0.02 |
| T5 + UMAP | t-STE | CCA | Region | Text + flavor | SVM | **0.04** |
| T5 + t-SNE | t-STE | ICP | Region | Text + flavor | SVM | 0.00 |
| T5 + t-SNE | t-STE | SNaCK | Region | Text + flavor | SVM | 0.01 |
| Random | | | Grape | | | 0.03 |
| T5 + UMAP | | | Grape | Text only | SVM | 0.02 |
| T5 + t-SNE | | | Grape | Text only | SVM | 0.03 |
| T5 + UMAP | MDS | CCA | Grape | Text + flavor | SVM | 0.03 |
| T5 + t-SNE | MDS | ICP | Grape | Text + flavor | SVM | 0.03 |
| T5 + t-SNE | MDS | CCA | Grape | Text + flavor | SVM | **0.05** |
| T5 + UMAP | t-STE | CCA | Grape | Text + flavor | SVM | 0.05 |
| T5 + t-SNE | t-STE | ICP | Grape | Text + flavor | SVM | 0.03 |
| T5 + t-SNE | t-STE | SNaCK | Grape | Text + flavor | SVM | 0.03 |
| Random | | | Alc % | | | 0.17 |
| T5 + UMAP | | | Alc % | Text only | SVM | 0.34 |
| T5 + t-SNE | | | Alc % | Text only | SVM | 0.21 |
| T5 + UMAP | MDS | CCA | Alc % | Text + flavor | SVM | 0.50 |
| T5 + t-SNE | MDS | ICP | Alc % | Text + flavor | SVM | 0.32 |
| T5 + t-SNE | MDS | CCA | Alc % | Text + flavor | SVM | **0.55** |
| T5 + UMAP | t-STE | CCA | Alc % | Text + flavor | SVM | 0.50 |
| T5 + t-SNE | t-STE | ICP | Alc % | Text + flavor | SVM | 0.32 |
| T5 + t-SNE | t-STE | SNaCK | Alc % | Text + flavor | SVM | 0.36 |
| Random | | | Price | | | 0.10 |
| T5 + UMAP | | | Price | Text only | SVM | 0.10 |
| T5 + t-SNE | | | Price | Text only | SVM | 0.18 |
| T5 + UMAP | MDS | CCA | Price | Text + flavor | SVM | 0.22 |
| T5 + t-SNE | MDS | ICP | Price | Text + flavor | SVM | 0.23 |
| T5 + t-SNE | MDS | CCA | Price | Text + flavor | SVM | **0.24** |
| T5 + UMAP | t-STE | CCA | Price | Text + flavor | SVM | 0.17 |
| T5 + t-SNE | t-STE | ICP | Price | Text + flavor | SVM | 0.22 |
| T5 + t-SNE | t-STE | SNaCK | Price | Text + flavor | SVM | 0.16 |
| Random | | | Rating | | | 0.25 |
| T5 + UMAP | | | Rating | Text only | SVM | 0.24 |
| T5 + t-SNE | | | Rating | Text only | SVM | 0.43 |
| T5 + UMAP | MDS | CCA | Rating | Text + flavor | SVM | 0.48 |
| T5 + t-SNE | MDS | ICP | Rating | Text + flavor | SVM | 0.43 |
| T5 + t-SNE | MDS | CCA | Rating | Text + flavor | SVM | 0.39 |
| T5 + UMAP | t-STE | CCA | Rating | Text + flavor | SVM | **0.53** |
| T5 + t-SNE | t-STE | ICP | Rating | Text + flavor | SVM | 0.43 |
| T5 + t-SNE | t-STE | SNaCK | Rating | Text + flavor | SVM | 0.37 |
| Random | | | Year | | | 0.08 |
| T5 + UMAP | | | Year | Text only | SVM | 0.06 |
| T5 + t-SNE | | | Year | Text only | SVM | 0.10 |
| T5 + UMAP | MDS | CCA | Year | Text + flavor | SVM | 0.12 |
| T5 + t-SNE | MDS | ICP | Year | Text + flavor | SVM | 0.10 |
| T5 + t-SNE | MDS | CCA | Year | Text + flavor | SVM | **0.12** |
| T5 + UMAP | t-STE | CCA | Year | Text + flavor | SVM | 0.11 |
| T5 + t-SNE | t-STE | ICP | Year | Text + flavor | SVM | 0.10 |
| T5 + t-SNE | t-STE | SNaCK | Year | Text + flavor | SVM | 0.09 |

Table 10: **ALBERT:** Classification results.

| Machine Kernel | Human Kernel | Combiner | Class | Modality | Pred | ACC ↑ |
|---|---|---|---|---|---|---|
| Random | | | Country | | | 0.13 |
| ALBERT + UMAP | | | Country | Text only | SVM | 0.09 |
| ALBERT + t-SNE | | | Country | Text only | SVM | 0.16 |
| ALBERT + UMAP | MDS | CCA | Country | Text + flavor | SVM | **0.20** |
| ALBERT + t-SNE | MDS | ICP | Country | Text + flavor | SVM | 0.10 |
| ALBERT + t-SNE | MDS | CCA | Country | Text + flavor | SVM | 0.19 |
| ALBERT + UMAP | t-STE | CCA | Country | Text + flavor | SVM | 0.14 |
| ALBERT + t-SNE | t-STE | ICP | Country | Text + flavor | SVM | 0.10 |
| ALBERT + t-SNE | t-STE | SNaCK | Country | Text + flavor | SVM | 0.12 |
| Random | | | Region | | | 0.02 |
| ALBERT + UMAP | | | Region | Text only | SVM | **0.03** |
| ALBERT + t-SNE | | | Region | Text only | SVM | 0.00 |
| ALBERT + UMAP | MDS | CCA | Region | Text + flavor | SVM | **0.03** |
| ALBERT + t-SNE | MDS | ICP | Region | Text + flavor | SVM | **0.03** |
| ALBERT + t-SNE | MDS | CCA | Region | Text + flavor | SVM | 0.02 |
| ALBERT + UMAP | t-STE | CCA | Region | Text + flavor | SVM | **0.03** |
| ALBERT + t-SNE | t-STE | ICP | Region | Text + flavor | SVM | **0.03** |
| ALBERT + t-SNE | t-STE | SNaCK | Region | Text + flavor | SVM | **0.03** |
| Random | | | Grape | | | 0.03 |
| ALBERT + UMAP | | | Grape | Text only | SVM | 0.0 |
| ALBERT + t-SNE | | | Grape | Text only | SVM | 0.0 |
| ALBERT + UMAP | MDS | CCA | Grape | Text + flavor | SVM | 0.02 |
| ALBERT + t-SNE | MDS | ICP | Grape | Text + flavor | SVM | **0.04** |
| ALBERT + t-SNE | MDS | CCA | Grape | Text + flavor | SVM | 0.02 |
| ALBERT + UMAP | t-STE | CCA | Grape | Text + flavor | SVM | 0.02 |
| ALBERT + t-SNE | t-STE | ICP | Grape | Text + flavor | SVM | 0.03 |
| ALBERT + t-SNE | t-STE | SNaCK | Grape | Text + flavor | SVM | 0.02 |
| Random | | | Alc % | | | 0.17 |
| ALBERT + UMAP | | | Alc % | Text only | SVM | 0.11 |
| ALBERT + t-SNE | | | Alc % | Text only | SVM | 0.24 |
| ALBERT + UMAP | MDS | CCA | Alc % | Text + flavor | SVM | **0.46** |
| ALBERT + t-SNE | MDS | ICP | Alc % | Text + flavor | SVM | 0.34 |
| ALBERT + t-SNE | MDS | CCA | Alc % | Text + flavor | SVM | **0.46** |
| ALBERT + UMAP | t-STE | CCA | Alc % | Text + flavor | SVM | 0.41 |
| ALBERT + t-SNE | t-STE | ICP | Alc % | Text + flavor | SVM | 0.33 |
| ALBERT + t-SNE | t-STE | SNaCK | Alc % | Text + flavor | SVM | 0.41 |
| Random | | | Price | | | 0.10 |
| ALBERT + UMAP | | | Price | Text only | SVM | 0.09 |
| ALBERT + t-SNE | | | Price | Text only | SVM | 0.17 |
| ALBERT + UMAP | MDS | CCA | Price | Text + flavor | SVM | **0.27** |
| ALBERT + t-SNE | MDS | ICP | Price | Text + flavor | SVM | 0.26 |
| ALBERT + t-SNE | MDS | CCA | Price | Text + flavor | SVM | 0.26 |
| ALBERT + UMAP | t-STE | CCA | Price | Text + flavor | SVM | 0.24 |
| ALBERT + t-SNE | t-STE | ICP | Price | Text + flavor | SVM | 0.26 |
| ALBERT + t-SNE | t-STE | SNaCK | Price | Text + flavor | SVM | 0.24 |
| Random | | | Rating | | | 0.25 |
| ALBERT + UMAP | | | Rating | Text only | SVM | 0.16 |
| ALBERT + t-SNE | | | Rating | Text only | SVM | 0.35 |
| ALBERT + UMAP | MDS | CCA | Rating | Text + flavor | SVM | 0.44 |
| ALBERT + t-SNE | MDS | ICP | Rating | Text + flavor | SVM | 0.33 |
| ALBERT + t-SNE | MDS | CCA | Rating | Text + flavor | SVM | **0.55** |
| ALBERT + UMAP | t-STE | CCA | Rating | Text + flavor | SVM | 0.39 |
| ALBERT + t-SNE | t-STE | ICP | Rating | Text + flavor | SVM | 0.33 |
| ALBERT + t-SNE | t-STE | SNaCK | Rating | Text + flavor | SVM | 0.39 |
| Random | | | Year | | | 0.08 |
| ALBERT + UMAP | | | Year | Text only | SVM | 0.09 |
| ALBERT + t-SNE | | | Year | Text only | SVM | 0.09 |
| ALBERT + UMAP | MDS | CCA | Year | Text + flavor | SVM | **0.17** |
| ALBERT + t-SNE | MDS | ICP | Year | Text + flavor | SVM | 0.08 |
| ALBERT + t-SNE | MDS | CCA | Year | Text + flavor | SVM | 0.13 |
| ALBERT + UMAP | t-STE | CCA | Year | Text + flavor | SVM | 0.12 |
| ALBERT + t-SNE | t-STE | ICP | Year | Text + flavor | SVM | 0.08 |
| ALBERT + t-SNE | t-STE | SNaCK | Year | Text + flavor | SVM | 0.12 |

Table 11: **BART:** Classification results.

| Machine Kernel | Human Kernel | Combiner | Class | Modality | Pred | ACC ↑ |
|---|---|---|---|---|---|---|
| Random | | | Country | | | 0.13 |
| BART + UMAP | | | Country | Text only | SVM | 0.06 |
| BART + t-SNE | | | Country | Text only | SVM | 0.12 |
| BART + UMAP | MDS | CCA | Country | Text + flavor | SVM | 0.16 |
| BART + t-SNE | MDS | ICP | Country | Text + flavor | SVM | 0.17 |
| BART + t-SNE | MDS | CCA | Country | Text + flavor | SVM | 0.15 |
| BART + UMAP | t-STE | CCA | Country | Text + flavor | SVM | **0.21** |
| BART + t-SNE | t-STE | ICP | Country | Text + flavor | SVM | 0.17 |
| BART + t-SNE | t-STE | SNaCK | Country | Text + flavor | SVM | 0.15 |
| Random | | | Region | | | **0.02** |
| BART + UMAP | | | Region | Text only | SVM | 0.00 |
| BART + t-SNE | | | Region | Text only | SVM | 0.00 |
| BART + UMAP | MDS | CCA | Region | Text + flavor | SVM | 0.00 |
| BART + t-SNE | MDS | ICP | Region | Text + flavor | SVM | 0.00 |
| BART + t-SNE | MDS | CCA | Region | Text + flavor | SVM | 0.00 |
| BART + UMAP | t-STE | CCA | Region | Text + flavor | SVM | 0.01 |
| BART + t-SNE | t-STE | ICP | Region | Text + flavor | SVM | 0.00 |
| BART + t-SNE | t-STE | SNaCK | Region | Text + flavor | SVM | 0.00 |
| Random | | | Grape | | | 0.03 |
| BART + UMAP | | | Grape | Text only | SVM | 0.01 |
| BART + t-SNE | | | Grape | Text only | SVM | 0.03 |
| BART + UMAP | MDS | CCA | Grape | Text + flavor | SVM | 0.03 |
| BART + t-SNE | MDS | ICP | Grape | Text + flavor | SVM | 0.01 |
| BART + t-SNE | MDS | CCA | Grape | Text + flavor | SVM | 0.03 |
| BART + UMAP | t-STE | CCA | Grape | Text + flavor | SVM | **0.06** |
| BART + t-SNE | t-STE | ICP | Grape | Text + flavor | SVM | 0.01 |
| BART + t-SNE | t-STE | SNaCK | Grape | Text + flavor | SVM | 0.00 |
| Random | | | Alc % | | | 0.17 |
| BART + UMAP | | | Alc % | Text only | SVM | 0.30 |
| BART + t-SNE | | | Alc % | Text only | SVM | 0.32 |
| BART + UMAP | MDS | CCA | Alc % | Text + flavor | SVM | 0.44 |
| BART + t-SNE | MDS | ICP | Alc % | Text + flavor | SVM | 0.19 |
| BART + t-SNE | MDS | CCA | Alc % | Text + flavor | SVM | **0.47** |
| BART + UMAP | t-STE | CCA | Alc % | Text + flavor | SVM | **0.47** |
| BART + t-SNE | t-STE | ICP | Alc % | Text + flavor | SVM | 0.19 |
| BART + t-SNE | t-STE | SNaCK | Alc % | Text + flavor | SVM | 0.39 |
| Random | | | Price | | | 0.10 |
| BART + UMAP | | | Price | Text only | SVM | 0.14 |
| BART + t-SNE | | | Price | Text only | SVM | 0.21 |
| BART + UMAP | MDS | CCA | Price | Text + flavor | SVM | **0.29** |
| BART + t-SNE | MDS | ICP | Price | Text + flavor | SVM | 0.12 |
| BART + t-SNE | MDS | CCA | Price | Text + flavor | SVM | 0.23 |
| BART + UMAP | t-STE | CCA | Price | Text + flavor | SVM | 0.20 |
| BART + t-SNE | t-STE | ICP | Price | Text + flavor | SVM | 0.12 |
| BART + t-SNE | t-STE | SNaCK | Price | Text + flavor | SVM | 0.13 |
| Random | | | Rating | | | 0.25 |
| BART + UMAP | | | Rating | Text only | SVM | 0.26 |
| BART + t-SNE | | | Rating | Text only | SVM | 0.39 |
| BART + UMAP | MDS | CCA | Rating | Text + flavor | SVM | 0.45 |
| BART + t-SNE | MDS | ICP | Rating | Text + flavor | SVM | 0.40 |
| BART + t-SNE | MDS | CCA | Rating | Text + flavor | SVM | 0.49 |
| BART + UMAP | t-STE | CCA | Rating | Text + flavor | SVM | **0.52** |
| BART + t-SNE | t-STE | ICP | Rating | Text + flavor | SVM | 0.40 |
| BART + t-SNE | t-STE | SNaCK | Rating | Text + flavor | SVM | 0.29 |
| Random | | | Year | | | 0.08 |
| BART + UMAP | | | Year | Text only | SVM | 0.08 |
| BART + t-SNE | | | Year | Text only | SVM | **0.13** |
| BART + UMAP | MDS | CCA | Year | Text + flavor | SVM | 0.10 |
| BART + t-SNE | MDS | ICP | Year | Text + flavor | SVM | 0.09 |
| BART + t-SNE | MDS | CCA | Year | Text + flavor | SVM | 0.10 |
| BART + UMAP | t-STE | CCA | Year | Text + flavor | SVM | **0.13** |
| BART + t-SNE | t-STE | ICP | Year | Text + flavor | SVM | 0.06 |
| BART + t-SNE | t-STE | SNaCK | Year | Text + flavor | SVM | 0.10 |

Table 12: **ViT:** Classification results.

| Machine Kernel | Human Kernel | Combiner | Class | Modality | Pred | ACC ↑ |
|---|---|---|---|---|---|---|
| Random | | | Country | | | 0.13 |
| ViT + UMAP | | | Country | Image only | SVM | 0.08 |
| ViT + t-SNE | | | Country | Image only | SVM | 0.16 |
| ViT + UMAP | MDS | CCA | Country | Image + flavor | SVM | 0.15 |
| ViT + t-SNE | MDS | ICP | Country | Image + flavor | SVM | 0.12 |
| ViT + t-SNE | MDS | CCA | Country | Image + flavor | SVM | **0.21** |
| ViT + UMAP | t-STE | CCA | Country | Image + flavor | SVM | 0.20 |
| ViT + t-SNE | t-STE | ICP | Country | Image + flavor | SVM | 0.07 |
| ViT + t-SNE | t-STE | SNaCK | Country | Image + flavor | SVM | 0.16 |
| Random | | | Region | | | 0.02 |
| ViT + UMAP | | | Region | Image only | SVM | 0.00 |
| ViT + t-SNE | | | Region | Image only | SVM | 0.01 |
| ViT + UMAP | MDS | CCA | Region | Image + flavor | SVM | **0.03** |
| ViT + t-SNE | MDS | ICP | Region | Image + flavor | SVM | 0.01 |
| ViT + t-SNE | MDS | CCA | Region | Image + flavor | SVM | **0.03** |
| ViT + UMAP | t-STE | CCA | Region | Image + flavor | SVM | 0.01 |
| ViT + t-SNE | t-STE | ICP | Region | Image + flavor | SVM | 0.00 |
| ViT + t-SNE | t-STE | SNaCK | Region | Image + flavor | SVM | 0.00 |
| Random | | | Grape | | | 0.03 |
| ViT + UMAP | | | Grape | Image only | SVM | 0.00 |
| ViT + t-SNE | | | Grape | Image only | SVM | 0.01 |
| ViT + UMAP | MDS | CCA | Grape | Image + flavor | SVM | **0.06** |
| ViT + t-SNE | MDS | ICP | Grape | Image + flavor | SVM | 0.00 |
| ViT + t-SNE | MDS | CCA | Grape | Image + flavor | SVM | 0.03 |
| ViT + UMAP | t-STE | CCA | Grape | Image + flavor | SVM | 0.03 |
| ViT + t-SNE | t-STE | ICP | Grape | Image + flavor | SVM | 0.00 |
| ViT + t-SNE | t-STE | SNaCK | Grape | Image + flavor | SVM | 0.00 |
| Random | | | Alc % | | | 0.17 |
| ViT + t-SNE | | | Alc % | Image only | SVM | 0.19 |
| ViT + UMAP | MDS | CCA | Alc % | Image only | SVM | 0.31 |
| ViT + t-SNE | MDS | ICP | Alc % | Image + flavor | SVM | 0.18 |
| ViT + t-SNE | MDS | CCA | Alc % | Image + flavor | SVM | **0.39** |
| ViT + UMAP | t-STE | CCA | Alc % | Image + flavor | SVM | 0.36 |
| ViT + t-SNE | t-STE | ICP | Alc % | Image + flavor | SVM | 0.11 |
| ViT + t-SNE | t-STE | SNaCK | Alc % | Image + flavor | SVM | 0.19 |
| Random | | | Price | | | 0.10 |
| ViT + UMAP | | | Price | Image only | SVM | 0.18 |
| ViT + t-SNE | | | Price | Image only | SVM | 0.21 |
| ViT + UMAP | MDS | CCA | Price | Image + flavor | SVM | **0.33** |
| ViT + t-SNE | MDS | ICP | Price | Image + flavor | SVM | 0.16 |
| ViT + t-SNE | MDS | CCA | Price | Image + flavor | SVM | 0.31 |
| ViT + UMAP | t-STE | CCA | Price | Image + flavor | SVM | 0.24 |
| ViT + t-SNE | t-STE | ICP | Price | Image + flavor | SVM | 0.17 |
| ViT + t-SNE | t-STE | SNaCK | Price | Image + flavor | SVM | 0.27 |
| Random | | | Rating | | | 0.25 |
| ViT + UMAP | | | Rating | Image only | SVM | 0.23 |
| ViT + t-SNE | | | Rating | Image only | SVM | 0.31 |
| ViT + UMAP | MDS | CCA | Rating | Image + flavor | SVM | 0.45 |
| ViT + t-SNE | MDS | ICP | Rating | Image + flavor | SVM | 0.31 |
| ViT + t-SNE | MDS | CCA | Rating | Image + flavor | SVM | 0.43 |
| ViT + UMAP | t-STE | CCA | Rating | Image + flavor | SVM | **0.58** |
| ViT + t-SNE | t-STE | ICP | Rating | Image + flavor | SVM | 0.31 |
| ViT + t-SNE | t-STE | SNaCK | Rating | Image + flavor | SVM | 0.32 |
| Random | | | Year | | | 0.08 |
| ViT + UMAP | | | Year | Image only | SVM | 0.06 |
| ViT + t-SNE | | | Year | Image only | SVM | 0.10 |
| ViT + UMAP | MDS | CCA | Year | Image + flavor | SVM | 0.10 |
| ViT + t-SNE | MDS | ICP | Year | Image + flavor | SVM | **0.14** |
| ViT + t-SNE | MDS | CCA | Year | Image + flavor | SVM | 0.09 |
| ViT + UMAP | t-STE | CCA | Year | Image + flavor | SVM | 0.08 |
| ViT + t-SNE | t-STE | ICP | Year | Image + flavor | SVM | 0.08 |
| ViT + t-SNE | t-STE | SNaCK | Year | Image + flavor | SVM | **0.14** |

Table 13: **ResNET:** Classification results.

| Machine Kernel | Human Kernel | Combiner | Class | Modality | Pred | ACC ↑ |
|---|---|---|---|---|---|---|
| Random | | | Country | | | 0.13 |
| ResNET + UMAP | | | Country | Image only | SVM | 0.10 |
| ResNET + t-SNE | | | Country | Image only | SVM | 0.13 |
| ResNET + UMAP | MDS | CCA | Country | Image + flavor | SVM | 0.23 |
| ResNET + t-SNE | MDS | ICP | Country | Image + flavor | SVM | 0.17 |
| ResNET + t-SNE | MDS | CCA | Country | Image + flavor | SVM | **0.24** |
| ResNET + UMAP | t-STE | CCA | Country | Image + flavor | SVM | 0.20 |
| ResNET + t-SNE | t-STE | ICP | Country | Image + flavor | SVM | 0.17 |
| ResNET + t-SNE | t-STE | SNaCK | Country | Image + flavor | SVM | 0.15 |
| Random | | | Region | | | **0.02** |
| ResNET + UMAP | | | Region | Image only | SVM | 0.00 |
| ResNET + t-SNE | | | Region | Image only | SVM | 0.00 |
| ResNET + UMAP | MDS | CCA | Region | Image + flavor | SVM | **0.02** |
| ResNET + t-SNE | MDS | ICP | Region | Image + flavor | SVM | 0.01 |
| ResNET + t-SNE | MDS | CCA | Region | Image + flavor | SVM | 0.01 |
| ResNET + UMAP | t-STE | CCA | Region | Image + flavor | SVM | 0.01 |
| ResNET + t-SNE | t-STE | ICP | Region | Image + flavor | SVM | 0.01 |
| ResNET + t-SNE | t-STE | SNaCK | Region | Image + flavor | SVM | 0.00 |
| Random | | | Grape | | | **0.03** |
| ResNET + UMAP | | | Grape | Image only | SVM | 0.00 |
| ResNET + t-SNE | | | Grape | Image only | SVM | 0.00 |
| ResNET + UMAP | MDS | CCA | Grape | Image + flavor | SVM | **0.03** |
| ResNET + t-SNE | MDS | ICP | Grape | Image + flavor | SVM | 0.00 |
| ResNET + t-SNE | MDS | CCA | Grape | Image + flavor | SVM | **0.03** |
| ResNET + UMAP | t-STE | CCA | Grape | Image + flavor | SVM | **0.03** |
| ResNET + t-SNE | t-STE | ICP | Grape | Image + flavor | SVM | 0.00 |
| ResNET + t-SNE | t-STE | SNaCK | Grape | Image + flavor | SVM | 0.00 |
| Random | | | Alc % | | | 0.17 |
| ResNET + UMAP | | | Alc % | Image only | SVM | 0.16 |
| ResNET + t-SNE | | | Alc % | Image only | SVM | 0.18 |
| ResNET + UMAP | MDS | CCA | Alc % | Image + flavor | SVM | 0.35 |
| ResNET + t-SNE | MDS | ICP | Alc % | Image + flavor | SVM | 0.14 |
| ResNET + t-SNE | MDS | CCA | Alc % | Image + flavor | SVM | **0.40** |
| ResNET + UMAP | t-STE | CCA | Alc % | Image + flavor | SVM | 0.36 |
| ResNET + t-SNE | t-STE | ICP | Alc % | Image + flavor | SVM | 0.14 |
| ResNET + t-SNE | t-STE | SNaCK | Alc % | Image + flavor | SVM | 0.19 |
| Random | | | Price | | | 0.10 |
| ResNET + UMAP | | | Price | Image only | SVM | 0.29 |
| ResNET + t-SNE | | | Price | Image only | SVM | 0.28 |
| ResNET + UMAP | MDS | CCA | Price | Image + flavor | SVM | **0.30** |
| ResNET + t-SNE | MDS | ICP | Price | Image + flavor | SVM | 0.29 |
| ResNET + t-SNE | MDS | CCA | Price | Image + flavor | SVM | **0.30** |
| ResNET + UMAP | t-STE | CCA | Price | Image + flavor | SVM | 0.29 |
| ResNET + t-SNE | t-STE | ICP | Price | Image + flavor | SVM | 0.29 |
| ResNET + t-SNE | t-STE | SNaCK | Price | Image + flavor | SVM | 0.28 |
| Random | | | Rating | | | 0.25 |
| ResNET + UMAP | | | Rating | Image only | SVM | 0.37 |
| ResNET + t-SNE | | | Rating | Image only | SVM | 0.34 |
| ResNET + UMAP | MDS | CCA | Rating | Image + flavor | SVM | 0.50 |
| ResNET + t-SNE | MDS | ICP | Rating | Image + flavor | SVM | 0.34 |
| ResNET + t-SNE | MDS | CCA | Rating | Image + flavor | SVM | 0.42 |
| ResNET + UMAP | t-STE | CCA | Rating | Image + flavor | SVM | **0.58** |
| ResNET + t-SNE | t-STE | ICP | Rating | Image + flavor | SVM | 0.34 |
| ResNET + t-SNE | t-STE | SNaCK | Rating | Image + flavor | SVM | 0.20 |
| Random | | | Year | | | 0.08 |
| ResNET + UMAP | | | Year | Image only | SVM | 0.08 |
| ResNET + t-SNE | | | Year | Image only | SVM | 0.10 |
| ResNET + UMAP | MDS | CCA | Year | Image + flavor | SVM | **0.11** |
| ResNET + t-SNE | MDS | ICP | Year | Image + flavor | SVM | 0.08 |
| ResNET + t-SNE | MDS | CCA | Year | Image + flavor | SVM | 0.09 |
| ResNET + UMAP | t-STE | CCA | Year | Image + flavor | SVM | 0.08 |
| ResNET + t-SNE | t-STE | ICP | Year | Image + flavor | SVM | 0.08 |
| ResNET + t-SNE | t-STE | SNaCK | Year | Image + flavor | SVM | 0.04 |

Table 14: **DeiT:** Classification results.

| Machine Kernel | Human Kernel | Combiner | Class | Modality | Pred | ACC ↑ |
|---|---|---|---|---|---|---|
| Random | | | Country | | | 0.13 |
| DeiT + UMAP | | | Country | Image only | SVM | 0.05 |
| DeiT + t-SNE | | | Country | Image only | SVM | 0.16 |
| DeiT + UMAP | MDS | CCA | Country | Image + flavor | SVM | **0.29** |
| DeiT + t-SNE | MDS | ICP | Country | Image + flavor | SVM | 0.12 |
| DeiT + t-SNE | MDS | CCA | Country | Image + flavor | SVM | 0.23 |
| DeiT + UMAP | t-STE | CCA | Country | Image + flavor | SVM | 0.26 |
| DeiT + t-SNE | t-STE | ICP | Country | Image + flavor | SVM | 0.13 |
| DeiT + t-SNE | t-STE | SNaCK | Country | Image + flavor | SVM | 0.12 |
| Random | | | Region | | | 0.02 |
| DeiT + UMAP | | | Region | Image only | SVM | 0.01 |
| DeiT + t-SNE | | | Region | Image only | SVM | 0.01 |
| DeiT + UMAP | MDS | CCA | Region | Image + flavor | SVM | **0.05** |
| DeiT + t-SNE | MDS | ICP | Region | Image + flavor | SVM | 0.01 |
| DeiT + t-SNE | MDS | CCA | Region | Image + flavor | SVM | 0.03 |
| DeiT + UMAP | t-STE | CCA | Region | Image + flavor | SVM | 0.02 |
| DeiT + t-SNE | t-STE | ICP | Region | Image + flavor | SVM | 0.01 |
| DeiT + t-SNE | t-STE | SNaCK | Region | Image + flavor | SVM | 0.0 |
| Random | | | Grape | | | 0.03 |
| DeiT + UMAP | | | Grape | Image only | SVM | 0.01 |
| DeiT + t-SNE | | | Grape | Image only | SVM | 0.01 |
| DeiT + UMAP | MDS | CCA | Grape | Image + flavor | SVM | **0.06** |
| DeiT + t-SNE | MDS | ICP | Grape | Image + flavor | SVM | 0.00 |
| DeiT + t-SNE | MDS | CCA | Grape | Image + flavor | SVM | **0.06** |
| DeiT + UMAP | t-STE | CCA | Grape | Image + flavor | SVM | 0.04 |
| DeiT + t-SNE | t-STE | ICP | Grape | Image + flavor | SVM | 0.0 |
| DeiT + t-SNE | t-STE | SNaCK | Grape | Image + flavor | SVM | 0.02 |
| Random | | | Alc % | | | 0.17 |
| DeiT + UMAP | | | Alc % | Image only | SVM | 0.13 |
| DeiT + t-SNE | | | Alc % | Image only | SVM | 0.19 |
| DeiT + UMAP | MDS | CCA | Alc % | Image + flavor | SVM | **0.39** |
| DeiT + t-SNE | MDS | ICP | Alc % | Image + flavor | SVM | 0.18 |
| DeiT + t-SNE | MDS | CCA | Alc % | Image + flavor | SVM | 0.33 |
| DeiT + UMAP | t-STE | CCA | Alc % | Image + flavor | SVM | **0.39** |
| DeiT + t-SNE | t-STE | ICP | Alc % | Image + flavor | SVM | 0.18 |
| DeiT + t-SNE | t-STE | SNaCK | Alc % | Image + flavor | SVM | 0.23 |
| Random | | | Price | | | 0.10 |
| DeiT + UMAP | | | Price | Image only | SVM | 0.21 |
| DeiT + t-SNE | | | Price | Image only | SVM | 0.21 |
| DeiT + UMAP | MDS | CCA | Price | Image + flavor | SVM | **0.38** |
| DeiT + t-SNE | MDS | ICP | Price | Image + flavor | SVM | 0.16 |
| DeiT + t-SNE | MDS | CCA | Price | Image + flavor | SVM | **0.38** |
| DeiT + UMAP | t-STE | CCA | Price | Image + flavor | SVM | 0.29 |
| DeiT + t-SNE | t-STE | ICP | Price | Image + flavor | SVM | 0.16 |
| DeiT + t-SNE | t-STE | SNaCK | Price | Image + flavor | SVM | 0.18 |
| Random | | | Rating | | | 0.25 |
| DeiT + UMAP | | | Rating | Image only | SVM | 0.29 |
| DeiT + t-SNE | | | Rating | Image only | SVM | 0.31 |
| DeiT + UMAP | MDS | CCA | Rating | Image + flavor | SVM | 0.32 |
| DeiT + t-SNE | MDS | ICP | Rating | Image + flavor | SVM | 0.31 |
| DeiT + t-SNE | MDS | CCA | Rating | Image + flavor | SVM | 0.44 |
| DeiT + UMAP | t-STE | CCA | Rating | Image + flavor | SVM | **0.49** |
| DeiT + t-SNE | t-STE | ICP | Rating | Image + flavor | SVM | 0.30 |
| DeiT + t-SNE | t-STE | SNaCK | Rating | Image + flavor | SVM | 0.28 |
| Random | | | Year | | | 0.08 |
| DeiT + UMAP | | | Year | Image only | SVM | 0.06 |
| DeiT + t-SNE | | | Year | Image only | SVM | 0.10 |
| DeiT + UMAP | MDS | CCA | Year | Image + flavor | SVM | 0.10 |
| DeiT + t-SNE | MDS | ICP | Year | Image + flavor | SVM | 0.14 |
| DeiT + t-SNE | MDS | CCA | Year | Image + flavor | SVM | 0.11 |
| DeiT + UMAP | t-STE | CCA | Year | Image + flavor | SVM | **0.15** |
| DeiT + t-SNE | t-STE | ICP | Year | Image + flavor | SVM | 0.14 |
| DeiT + t-SNE | t-STE | SNaCK | Year | Image + flavor | SVM | 0.12 |

Table 15: **CLIP (Image Encoder):** Classification results.

| Machine Kernel | Human Kernel | Combiner | Class | Modality | Pred | ACC ↑ |
|---|---|---|---|---|---|---|
| Random | | | Country | | | 0.13 |
| CLIP + UMAP | | | Country | Image only | SVM | 0.08 |
| CLIP + t-SNE | | | Country | Image only | SVM | 0.05 |
| CLIP + UMAP | MDS | CCA | Country | Image + flavor | SVM | 0.21 |
| CLIP + t-SNE | MDS | ICP | Country | Image + flavor | SVM | 0.08 |
| CLIP + t-SNE | MDS | CCA | Country | Image + flavor | SVM | 0.24 |
| CLIP + UMAP | t-STE | CCA | Country | Image + flavor | SVM | **0.57** |
| CLIP + t-SNE | t-STE | ICP | Country | Image + flavor | SVM | 0.53 |
| CLIP + t-SNE | t-STE | SNaCK | Country | Image + flavor | SVM | 0.48 |
| Random | | | Region | | | 0.02 |
| CLIP + UMAP | | | Region | Image only | SVM | 0.00 |
| CLIP + t-SNE | | | Region | Image only | SVM | 0.00 |
| CLIP + UMAP | MDS | CCA | Region | Image + flavor | SVM | 0.02 |
| CLIP + t-SNE | MDS | ICP | Region | Image + flavor | SVM | 0.00 |
| CLIP + t-SNE | MDS | CCA | Region | Image + flavor | SVM | 0.00 |
| CLIP + UMAP | t-STE | CCA | Region | Image + flavor | SVM | **0.04** |
| CLIP + t-SNE | t-STE | ICP | Region | Image + flavor | SVM | 0.03 |
| CLIP + t-SNE | t-STE | SNaCK | Region | Image + flavor | SVM | **0.04** |
| Random | | | Grape | | | 0.03 |
| CLIP + UMAP | | | Grape | Image only | SVM | 0.00 |
| CLIP + t-SNE | | | Grape | Image only | SVM | 0.00 |
| CLIP + UMAP | MDS | CCA | Grape | Image + flavor | SVM | 0.07 |
| CLIP + t-SNE | MDS | ICP | Grape | Image + flavor | SVM | 0.00 |
| CLIP + t-SNE | MDS | CCA | Grape | Image + flavor | SVM | 0.05 |
| CLIP + UMAP | t-STE | CCA | Grape | Image + flavor | SVM | **0.15** |
| CLIP + t-SNE | t-STE | ICP | Grape | Image + flavor | SVM | 0.09 |
| CLIP + t-SNE | t-STE | SNaCK | Grape | Image + flavor | SVM | 0.10 |
| Random | | | Alc % | | | 0.17 |
| CLIP + UMAP | | | Alc % | Image only | SVM | 0.11 |
| CLIP + t-SNE | | | Alc % | Image only | SVM | 0.11 |
| CLIP + UMAP | MDS | CCA | Alc % | Image + flavor | SVM | 0.42 |
| CLIP + t-SNE | MDS | ICP | Alc % | Image + flavor | SVM | 0.18 |
| CLIP + t-SNE | MDS | CCA | Alc % | Image + flavor | SVM | 0.44 |
| CLIP + UMAP | t-STE | CCA | Alc % | Image + flavor | SVM | **0.46** |
| CLIP + t-SNE | t-STE | ICP | Alc % | Image + flavor | SVM | 0.35 |
| CLIP + t-SNE | t-STE | SNaCK | Alc % | Image + flavor | SVM | 0.31 |
| Random | | | Price | | | 0.10 |
| CLIP + UMAP | | | Price | Image only | SVM | 0.20 |
| CLIP + t-SNE | | | Price | Image only | SVM | 0.16 |
| CLIP + UMAP | MDS | CCA | Price | Image + flavor | SVM | 0.28 |
| CLIP + t-SNE | MDS | ICP | Price | Image + flavor | SVM | 0.20 |
| CLIP + t-SNE | MDS | CCA | Price | Image + flavor | SVM | **0.30** |
| CLIP + UMAP | t-STE | CCA | Price | Image + flavor | SVM | 0.29 |
| CLIP + t-SNE | t-STE | ICP | Price | Image + flavor | SVM | 0.09 |
| CLIP + t-SNE | t-STE | SNaCK | Price | Image + flavor | SVM | 0.16 |
| Random | | | Rating | | | 0.25 |
| CLIP + UMAP | | | Rating | Image only | SVM | 0.15 |
| CLIP + t-SNE | | | Rating | Image only | SVM | 0.12 |
| CLIP + UMAP | MDS | CCA | Rating | Image + flavor | SVM | 0.36 |
| CLIP + t-SNE | MDS | ICP | Rating | Image + flavor | SVM | 0.20 |
| CLIP + t-SNE | MDS | CCA | Rating | Image + flavor | SVM | 0.39 |
| CLIP + UMAP | t-STE | CCA | Rating | Image + flavor | SVM | **0.47** |
| CLIP + t-SNE | t-STE | ICP | Rating | Image + flavor | SVM | 0.28 |
| CLIP + t-SNE | t-STE | SNaCK | Rating | Image + flavor | SVM | 0.42 |
| Random | | | Year | | | 0.08 |
| CLIP + UMAP | | | Year | Image only | SVM | 0.37 |
| CLIP + t-SNE | | | Year | Image only | SVM | 0.30 |
| CLIP + UMAP | MDS | CCA | Year | Image + flavor | SVM | **0.38** |
| CLIP + t-SNE | MDS | ICP | Year | Image + flavor | SVM | 0.29 |
| CLIP + t-SNE | MDS | CCA | Year | Image + flavor | SVM | 0.20 |
| CLIP + UMAP | t-STE | CCA | Year | Image + flavor | SVM | 0.12 |
| CLIP + t-SNE | t-STE | ICP | Year | Image + flavor | SVM | 0.12 |
| CLIP + t-SNE | t-STE | SNaCK | Year | Image + flavor | SVM | 0.11 |

Table 16: **CLIP (Image and Text Encoder):** Classification results.

| Machine Kernel | Human Kernel | Combiner | Class | Modality | Pred | ACC ↑ |
|---|---|---|---|---|---|---|
| Random | | | Country | | | 0.13 |
| CLIP + UMAP | | | Country | Image + text | SVM | 0.38 |
| CLIP + t-SNE | | | Country | Image + text | SVM | 0.48 |
| CLIP + UMAP | MDS | CCA | Country | Image + text + flavor | SVM | 0.44 |
| CLIP + t-SNE | MDS | ICP | Country | Image + text + flavor | SVM | **0.53** |
| CLIP + t-SNE | MDS | CCA | Country | Image + text + flavor | SVM | 0.45 |
| CLIP + UMAP | t-STE | CCA | Country | Image + text + flavor | SVM | 0.38 |
| CLIP + t-SNE | t-STE | ICP | Country | Image + text + flavor | SVM | **0.53** |
| CLIP + t-SNE | t-STE | SNaCK | Country | Image + text + flavor | SVM | 0.48 |
| Random | | | Region | | | 0.02 |
| CLIP + UMAP | | | Region | Image + text | SVM | 0.06 |
| CLIP + t-SNE | | | Region | Image + text | SVM | 0.04 |
| CLIP + UMAP | MDS | CCA | Region | Image + text + flavor | SVM | **0.07** |
| CLIP + t-SNE | MDS | ICP | Region | Image + text + flavor | SVM | 0.03 |
| CLIP + t-SNE | MDS | CCA | Region | Image + text + flavor | SVM | 0.06 |
| CLIP + UMAP | t-STE | CCA | Region | Image + text + flavor | SVM | 0.00 |
| CLIP + t-SNE | t-STE | ICP | Region | Image + text + flavor | SVM | 0.03 |
| CLIP + t-SNE | t-STE | SNaCK | Region | Image + text + flavor | SVM | 0.04 |
| Random | | | Grape | | | 0.03 |
| CLIP + UMAP | | | Grape | Image + text | SVM | 0.07 |
| CLIP + t-SNE | | | Grape | Image + text | SVM | 0.10 |
| CLIP + UMAP | MDS | CCA | Grape | Image + text + flavor | SVM | 0.06 |
| CLIP + t-SNE | MDS | ICP | Grape | Image + text + flavor | SVM | 0.09 |
| CLIP + t-SNE | MDS | CCA | Grape | Image + text + flavor | SVM | 0.06 |
| CLIP + UMAP | t-STE | CCA | Grape | Image + text + flavor | SVM | 0.07 |
| CLIP + t-SNE | t-STE | ICP | Grape | Image + text + flavor | SVM | 0.09 |
| CLIP + t-SNE | t-STE | SNaCK | Grape | Image + text + flavor | SVM | **0.10** |
| Random | | | Alc % | | | 0.17 |
| CLIP + UMAP | | | Alc % | Image + text | SVM | 0.09 |
| CLIP + t-SNE | | | Alc % | Image + text | SVM | 0.30 |
| CLIP + UMAP | MDS | CCA | Alc % | Image + text + flavor | SVM | **0.53** |
| CLIP + t-SNE | MDS | ICP | Alc % | Image + text + flavor | SVM | 0.35 |
| CLIP + t-SNE | MDS | CCA | Alc % | Image + text + flavor | SVM | **0.53** |
| CLIP + UMAP | t-STE | CCA | Alc % | Image + text + flavor | SVM | 0.43 |
| CLIP + t-SNE | t-STE | ICP | Alc % | Image + text + flavor | SVM | 0.35 |
| CLIP + t-SNE | t-STE | SNaCK | Alc % | Image + text + flavor | SVM | 0.31 |
| Random | | | Price | | | 0.10 |
| CLIP + UMAP | | | Price | Image + text | SVM | 0.18 |
| CLIP + t-SNE | | | Price | Image + text | SVM | 0.18 |
| CLIP + UMAP | MDS | CCA | Price | Image + text + flavor | SVM | **0.33** |
| CLIP + t-SNE | MDS | ICP | Price | Image + text + flavor | SVM | 0.09 |
| CLIP + t-SNE | MDS | CCA | Price | Image + text + flavor | SVM | 0.30 |
| CLIP + UMAP | t-STE | CCA | Price | Image + text + flavor | SVM | 0.32 |
| CLIP + t-SNE | t-STE | ICP | Price | Image + text + flavor | SVM | 0.09 |
| CLIP + t-SNE | t-STE | SNaCK | Price | Image + text + flavor | SVM | 0.15 |
| Random | | | Rating | | | 0.25 |
| CLIP + UMAP | | | Rating | Image + text | SVM | 0.23 |
| CLIP + t-SNE | | | Rating | Image + text | SVM | 0.33 |
| CLIP + UMAP | MDS | CCA | Rating | Image + text + flavor | SVM | 0.40 |
| CLIP + t-SNE | MDS | ICP | Rating | Image + text + flavor | SVM | 0.29 |
| CLIP + t-SNE | MDS | CCA | Rating | Image + text + flavor | SVM | 0.42 |
| CLIP + UMAP | t-STE | CCA | Rating | Image + text + flavor | SVM | **0.45** |
| CLIP + t-SNE | t-STE | ICP | Rating | Image + text + flavor | SVM | 0.29 |
| CLIP + t-SNE | t-STE | SNaCK | Rating | Image + text + flavor | SVM | 0.42 |
| Random | | | Year | | | 0.08 |
| CLIP + UMAP | | | Year | Image + text | SVM | 0.07 |
| CLIP + t-SNE | | | Year | Image + text | SVM | 0.09 |
| CLIP + UMAP | MDS | CCA | Year | Image + text + flavor | SVM | 0.10 |
| CLIP + t-SNE | MDS | ICP | Year | Image + text + flavor | SVM | 0.12 |
| CLIP + t-SNE | MDS | CCA | Year | Image + text + flavor | SVM | **0.17** |
| CLIP + UMAP | t-STE | CCA | Year | Image + text + flavor | SVM | 0.16 |
| CLIP + t-SNE | t-STE | ICP | Year | Image + text + flavor | SVM | 0.12 |
| CLIP + t-SNE | t-STE | SNaCK | Year | Image + text + flavor | SVM | 0.11 |

## G    Ethical approval

The original ethical approval is shown in Figure 7 English translation of the ethical approval can be found in section G.1.

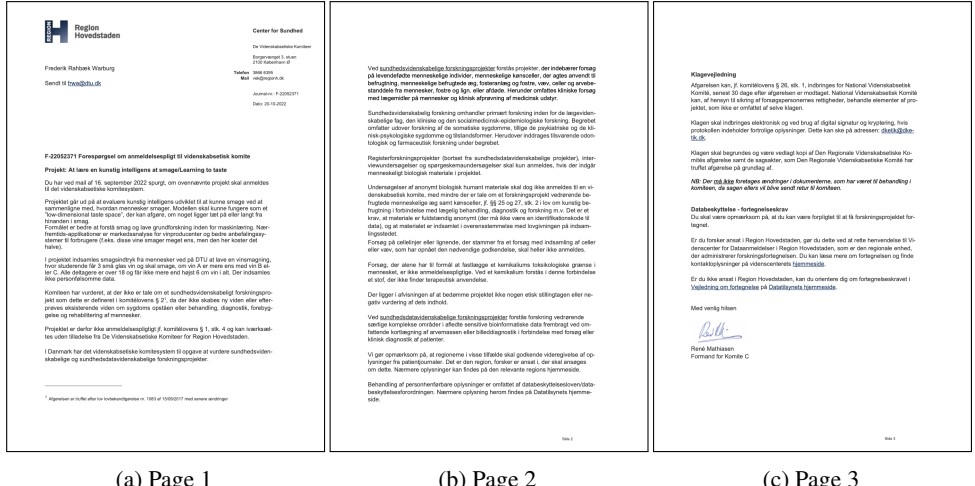

(a) Page 1                                (b) Page 2                                (c) Page 3

Figure 7: Ethical Approval (in Danish).

### G.1    English translation

**F-22052371 Inquiry Regarding Reporting Obligations to the Ethical Scientific Committee**

**Project: Learning to Taste**

You have asked via email on September 16, 2022, if the above-mentioned project must be reported to the Ethical Scientific Committee. The project involves evaluating an artificial intelligence developed to mimic the human ability to taste, comparing it with the way humans experience flavors. The model should function as a "low-dimensional taste space", which can determine whether something is close to or far from each other in terms of taste.

The aim is to better understand taste and conduct basic research in machine learning. Near-future applications include market analysis for wine producers and improved recommendation systems for consumers (e.g., these wines taste very similar, but this one costs half as much).

In the project, taste impressions from humans are collected by conducting a wine tasting at DTU, where students are given three small glasses of wine to taste whether wine A is more similar to wine B or C. All participants are over 18 and receive no more than a maximum of 6 cl of wine in total. No sensitive personal data is collected.

The committee has assessed that this is not a health science research project as defined in the committee law's section 21, as it does not create new knowledge or test existing knowledge about disease onset or treatment, diagnostics, prevention, and rehabilitation of humans.

Therefore, the project is not subject to reporting according to the committee law's section 1, paragraph 4 and can be implemented without permission from the Ethical Scientific Committees for the Capital Region of Denmark.

In Denmark, the task of the Ethical Scientific Committee system is to assess health science and health data science research projects.

Health science research projects refer to experiments involving live-born human individuals, human gametes intended for fertilization, human fertilized eggs, embryonic and fetal tissues, cells, and

hereditary components from humans, fetuses, and the like, or deceased individuals. This includes clinical trials with drugs on humans and clinical testing of medical equipment.

Health science research primarily covers research in the field of medical science, clinical, and social-medical epidemiological research. In addition to research on somatic diseases, the term also encompasses psychiatric and clinical-psychological diseases and conditions. Correspondingly, dental and pharmaceutical research are included under the term.

Registered research projects (except for health data science projects), interviews, and questionnaire surveys only need to be reported if human biological material is included in the project. However, investigations of anonymous biological human material do not need to be reported to an ethical scientific committee unless the research project relates to fertilized human eggs and sex cells, cf. sections 25 and 27, paragraph 2 in the Act on Artificial Fertilization in connection with medical treatment, diagnosis, and research. It is a requirement that the material is completely anonymous (there must not be an identification code for data), and that the material is collected in accordance with the law at the collection site.

Experiments on cell lines or similar originating from an experiment collecting cells or tissue, which has received the necessary approval, also do not need to be reported. Experiments that aim solely to determine a chemical's toxicological limit in humans do not need to be reported. In this context, a chemical is understood to mean a substance that does not find therapeutic use.

The rejection to review the project does not imply an ethical stance or negative assessment of its content.

Health data science research projects refer to research concerning particular complex areas of derived sensitive bio-information data produced by comprehensive mapping of the genetic mass or imaging diagnostics in connection with experiments or clinical diagnostics of patients.

We note that in certain cases, the regions must approve the disclosure of information from patient records. The region in which the researcher is employed must be applied to for this. More information can be found on the relevant region's website.

The processing of identifiable personal information is subject to the Data Protection Act/Data Protection Regulation. More information about this can be found on the Danish Data Protection Agency's website.

According to section 26, paragraph 1 of the Committee Act, the decision can be appealed to the National Ethical Scientific Committee no later than 30 days after the decision has been received. The National Ethical Scientific Committee may, for the sake of safeguarding the rights of the test subjects, handle aspects of the project not covered by the appeal itself.

Appeals must be filed electronically and using a digital signature and encryption if the protocol contains confidential information. This can be done at the address: dketik@dke-tik.dk.

The appeal must be justified and accompanied by a copy of the decision of the Regional Ethical Scientific Committee and the case documents on which the Regional Ethical Scientific Committee has made its decision.

**Note:** No changes should be made to the documents that have been reviewed by the committee, otherwise, the case will be returned to the committee.

**Data Protection - Registry Requirement**

Please note that you may be required to register the research project.

If you are a researcher employed in the Capital Region, you do this by contacting the Knowledge Center for Data Reviews in the Capital Region, which is the regional unit that administers the research registry. You can read more about the registry and find contact information on the knowledge center's website.

If you are not employed in the Capital Region, you can learn about the registry requirement in the Guide to the Registry on the Data Inspectorate's website.

Best regards,
René Mathiasen
Chairman of Committee C

# H Datasheet

## H.1 Motivation

**For what purpose was the dataset created?**

**Answer:** The dataset was created to bridge the gap between food science and machine learning communities and introduce flavor as a modality in multimodal models.

**Who created the dataset (e.g., which team, research group) and on behalf of which entity (e.g., company, institution, organization)?**

**Answer:** Eight researchers at the Technical University of Denmark, University of Copenhagen, Vivino and California Institute of Technology have created the dataset: Thoranna Bender, Simon Moe Søresen, Alireza Kashani, Kristjan Eldjarn Hjorleifsson, Grethe Hyldig, Søren Hauberg and Frederik Warburg.

**Who funded the creation of the dataset?**

**Answer:** The dataset is funded in part by The Danish Data Science Academy (DDSA) and the Pioneer Centre for AI (DNRF grant number P1).

**Any other comments?**

**Answer:** No.

## H.2 Composition

**What do the instances that comprise the dataset represent (e.g., documents, photos, people, countries)?**

**Answer:** Each instance is an image of a wine bottle, a review about the wine, position of the wines on napping papers and attributes (grape, country, region, alcohol %, price and rating).

**How many instances are there in total (of each type, if appropriate)?**

**Answer:** 897k images, 824k reviews of 350k vintages, around 5% of which are also associated with year, region, rating, alcohol percentage, and grape composition. In addition there are over 5k annotated pairwise flavor distances for 108 of the wines.

**Does the dataset contain all possible instances or is it a sample (not necessarily random) of instances from a larger set?**

**Answer:** The provided images, reviews and attributes are sampled from Vivino's database. The provided flavor annotations are provided in full for the 108 wines they exist for.

**What data does each instance consist of?**

**Answer:** The images are .jpg files, the reviews are unprocessed text, the attributes are either numerical or categorical fields and the flavor annotations are numerical x-axis and y-axix position annotations.

**Is there a label or target associated with each instance?**

**Answer:** No, but attributes can be used as targets as shown in section .

**Is any information missing from individual instances?**

**Answer:** Yes, the attributes are available for approximately 5% of the dataset and the flavor annotations are available for 108 vintages in the dataset.

**Are relationships between individual instances made explicit (e.g., users' movie ratings, social network links)?**

**Answer:** Yes, participant ID's are mappable to flavor annotations by using the values in the session_round_name, experiment_round and experiment_no fields in participants.csv and napping.csv.

**Are there recommended data splits (e.g., training, development/validation, testing)?**

**Answer:** No.

**Are there any errors, sources of noise, or redundancies in the dataset?**

**Answer:** No.

**Is the dataset self-contained, or does it link to or otherwise rely on external resources (e.g., websites, tweets, other datasets)?**

**Answer:** The data is self-contained.

**Does the dataset contain data that might be considered confidential (e.g., data that is protected by legal privilege or by doctor-patient confidentiality, data that includes the content of individuals' non-public communications)?**

**Answer:** No.

**Does the dataset contain data that, if viewed directly, might be offensive, insulting, threatening, or might otherwise cause anxiety?**

**Answer:** No.

**Does the dataset relate to people?**

**Answer:** Yes, but indirectly. Reviews, images and flavor annotations could provide some indirect information about the people annotating them (such as language used in reviews or background in images) but no attributes containing specific information about the people (such as gender, country, age etc.) exists in the dataset.

**Does the dataset identify any subpopulations (e.g., by age, gender)?**

**Answer:** No.

**Is it possible to identify individuals (i.e., one or more natural persons), either directly or indirectly (i.e., in combination with other data) from the dataset?**

**Answer:** No.

**Does the dataset contain data that might be considered sensitive in any way (e.g., data that reveals racial or ethnic origins, sexual orientations, religious beliefs, political opinions or union memberships, or locations; financial or health data; biometric or genetic data; forms of government identification, such as social security numbers; criminal history)?**

**Answer:** No.

**Any other comments?**

**Answer:** No.

### H.3 Collection process

**How was the data associated with each instance acquired?**

**Answer:** The flavor data was reported by subjects using the Napping method. The images, reviews and attributes were fetched from the Vivino platform. The flavor data was verified by a human manually checking the correctness of the algorithms annotating the napping papers. The attributes

have been verified by a human to correctly represent the information about individual vintages available on the Vivino platform.

**What mechanisms or procedures were used to collect the data (e.g., hardware apparatus or sensor, manual human curation, software program, software API)?**

**Answer:** Manual human curation and information fetched from Vivino's databases.

**If the dataset is a sample from a larger set, what was the sampling strategy (e.g., deterministic, probabilistic with specific sampling probabilities)?**

**Answer:** Not applicable.

**Who was involved in the data collection process (e.g., students, crowdworkers, contractors) and how were they compensated (e.g., how much were crowdworkers paid)?**

**Answer:** Crowd-workers that volunteered their time annotated the flavor distances. Alireza Kashani provided the image- and review data on behalf of Vivino. Attributes for the wines were collected from the Vivino platform.

**Over what timeframe was the data collected?**

**Answer:** The data was collected over the timeframe of June 2022 to May 2023.

**Were any ethical review processes conducted (e.g., by an institutional review board)?**

**Answer:** Yes, the ethical approval is provided in G.

**Does the dataset relate to people?**

**Answer:** Yes.

**Did you collect the data from the individuals in question directly, or obtain it via third parties or other sources (e.g., websites)?**

**Answer:** Obtained from the individuals directly.

**Were the individuals in question notified about the data collection?**

**Answer:** Yes.

**Did the individuals in question consent to the collection and use of their data?**

**Answer:** Yes.

**If consent was obtained, were the consenting individuals provided with a mechanism to revoke their consent in the future or for certain uses?**

**Answer:** No, this was not considered necessary, as the data can not be traced back to individuals.

**Has an analysis of the potential impact of the dataset and its use on data subjects (e.g., a data protection impact analysis) been conducted?**

**Answer:** No.

**Any other comments?**

**Answer:** No.

### H.4 Preprocessing/cleaning/labeling

**Was any preprocessing/cleaning/labeling of the data done (e.g., discretization or bucketing, tokenization, part-of-speech tagging, SIFT feature extraction, removal of instances, processing of missing values)?**

**Answer:** Yes, flavor annotation sample sheets from crowd-workers were digitized, by using the Harris corner detector [Harris et al., 1988] to find the corners of the paper and a homographic projection to obtain an aligned top-down view of the paper. The images were mapped into HSV color

space and a threshold filter applied to find the different colored stickers that the participant used to represent the wines. Having identified the location, we provide the Euclidean pixel-wise distance between all pairs of points in the dataset.

**Was the "raw" data saved in addition to the preprocessed/cleaned/labeled data (e.g., to support unanticipated future uses)?**

**Answer:** No, the sample sheets themselves were deemed to contain no information in addition to the pairwise distances provided.

**Is the software used to preprocess/clean/label the instances available?**

**Answer:** Yes, the preprocessing software is available at https://github.com/thoranna/learning_to_taste.

**Any other comments?**

**Answer:** No.

### H.5 Uses

**Has the dataset been used for any tasks already?**

**Answer:** Yes, the dataset has been used to classify different wines according to the attributes provided in the dataset.

**Is there a repository that links to any or all papers or systems that use the dataset?**

**Answer:** Yes, the analysis performed is available at https://github.com/thoranna/learning_to_taste.

**What (other) tasks could the dataset be used for?**

**Answer:** The dataset could be used for analyzing how similar different peoples' sense of taste is. It could also be used to identify wines that taste similar, but are available at different price points.

**Is there anything about the composition of the dataset or the way it was collected and preprocessed/cleaned/labeled that might impact future uses?**

**Answer:** Not to the authors' knowledge.

**Are there tasks for which the dataset should not be used?**

**Answer:** No.

**Any other comments?**

**Answer:** No.

### H.6 Distribution

**Will the dataset be distributed to third parties outside of the entity (e.g., company, institution, organization) on behalf of which the dataset was created?**

**Answer:** Yes, the dataset will be freely available to everyone.

**How will the dataset will be distributed (e.g., tarball on website, API, GitHub)?**

**Answer:** Tarball on website.

**When will the dataset be distributed?**

**Answer:** The dataset is freely availble as of June 12, 2023.

**Will the dataset be distributed under a copyright or other intellectual property (IP) license, and/or under applicable terms of use (ToU)?**

**Answer:** The dataset is available under Creative Commons Attribution 4.0 International License.

**Have any third parties imposed IP-based or other restrictions on the data associated with the instances?**

**Answer:** No.

**Do any export controls or other regulatory restrictions apply to the dataset or to individual instances?**

**Answer:** No.

**Any other comments?**

**Answer:** No.

### H.7 Maintenance

**Who is supporting/hosting/maintaining the dataset? How can the owner/curator/manager of the dataset be contacted (e.g., email address)?**

**Answer:** The maintainer of the dataset is Frederik Warburg (frewar1905@gmail.com)

**Is there an erratum?**

**Answer:** No.

**Will the dataset be updated (e.g., to correct labeling errors, add new instances, delete instances)?**

**Answer:** No.

**If the dataset relates to people, are there applicable limits on the retention of the data associated with the instances (e.g., were individuals in question told that their data would be retained for a fixed period of time and then deleted)?**

**Answer:** No.

**Will older versions of the dataset continue to be supported/hosted/maintained?**

**Answer:** Yes.

**If others want to extend/augment/build on/contribute to the dataset, is there a mechanism for them to do so?**

**Answer:** No, this will be resolved on a case-by-case basis, as the nature of the dataset requires data collection events for expansion.

**Any other comments?**

**Answer:** No.