# OpenReview forum: "Learning to Taste: A Multimodal Wine Dataset"
_NeurIPS.cc/2023/Track/Datasets_and_Benchmarks — NeurIPS 2023 Datasets and Benchmarks Poster_

### Official Review · Reviewer_N3w8 · 2023-07-19
**Modeling flavor perception is a breakthrough attempt.**

**Rating:** 6
**Confidence:** 4
**Correctness:** Yes.
**Clarity:** Yes.

**Strengths:**

- The dataset is unique and highly relevant, as flavor is an important but (extremely) understudied modality for multimodal deep learning. Modeling flavor perception could enable many applications in food science, consumer products, etc.

- The flavor annotation methodology using Napping is novel for machine learning and provides more nuanced human flavor similarity judgments than prior approaches like image triplets. The large scale of the study (256 people, 5K annotations) is impressive.

- Multimodal modeling with vision, language, and human flavor annotations is compelling and well-motivated. The proposed FEAST method for aligning representations is simple but effective.

- The comprehensive experiments cover many encoder, dimensionality reduction, and fusion options. The gains from multimodal learning validate the utility of the dataset.

- The dataset website is designed to look more fancy than the paper writing.

**Additional Feedback:**

N.A.

**Documentation:**

Yes.

**Ethics:**

N.A.

**Limitations:**

Yes.

**Opportunities For Improvement:**

- The flavor representations learned by FEAST are embedded spaces rather than natural language descriptions. As you noted, this could limit interpretability for humans. Adding natural language generation to explain the flavor spaces would be an interesting direction.

- Prediction accuracy on attributes is quite low, only around 20-30%, suggesting the learned representations may not yet capture flavors well enough for real-world use. More work is likely needed to improve the flavor modeling and dataset quality.

- The writing could be tightened up - for example, the formatting of the figures and flavor annotation definitions could be cleaner and more professional. As an initial dataset paper, the presentation could be polished further.

- The dataset currently focuses only on wine in Western Culture. Expanding to more drink types would increase culture diversity.

- The potential biases in the Vivino reviews (amateur enthusiasts) could be briefly discussed.

**Relation To Prior Work:**

Yes.

**Summary And Contributions:**

The authors present WineSensed, a new multimodal dataset for modeling flavor perception consisting of wine images, text reviews, metadata, and human flavor similarity annotations. The key contributions are:

- A large-scale dataset with 897k images, 824k text reviews, metadata for 350k wines, and 5k human flavor similarity annotations obtained from wine tasting experiments.

- A novel "Napping" annotation methodology to collect fine-grained human flavor similarities, allowing more nuanced flavor modeling compared to coarse attributes or classifications.

- A multimodal embedding method, FEAST, to align semantic text/image representations from CLIP with human flavor similarities using NMDS and CCA.

- Experiments demonstrating that combining modalities with FEAST improves performance on both coarse (attributes) and fine-grained (human flavor similarity) flavor prediction tasks.

- The introduction of flavor as an additional perceptive modality for representation learning, similar to vision, language, audio, etc. This provides a path towards more grounded ML models.

Overall, this is a novel dataset and interesting step towards modeling the complex human perception of flavor using a multimodal approach. The sensory evaluation methodology and size of the human annotations are impressive. FEAST provides a simple but effective way to align machine and human representations of flavor.

---

> ### Author Response · Authors · 2023-08-10
>
> >The flavor representations learned by FEAST are embedded spaces rather than natural language descriptions. As you noted, this could limit interpretability for humans. Adding natural language generation to explain the flavor spaces would be an interesting direction
>
> We thank the reviewer for the ideas, and we agree that other models than those employed in the manuscript might yield better results. Our main contribution is the data, which will enable such analysis. We see many interesting avenues for future exploration, such as decoding the latent space into “canonical wine labels” or “canonical text descriptions” and hope the community will adapt and use the dataset to help us explore these interesting possibilities that the data presents.
>
> >Prediction accuracy on attributes is quite low, only around 20-30%, suggesting the learned representation may not yet capture flavors well enough for real-world use. More work is likely needed to improve the flavor modeling and dataset quality.
>
> We thank the reviewer for raising this concern. It is not uncommon for benchmarks published along with novel datasets to have room for improvements. Compare for instance the 17.3% (vs. 1/200 random) benchmarked accuracy of the CUB200 dataset upon release (https://authors.library.caltech.edu/27452/1/CUB_200_2011.pdf) to the 93% accuracy achievable on the same dataset today (https://paperswithcode.com/sota/fine-grained-image-classification-on-cub-200). Part of the excitement of a novel dataset is that there is usually room for improvement on the analysis and modeling front. It would be worse if the dataset was already saturated.
>
> >The writing could be tightened up - for example, the formatting of the figures and flavor annotation definitions could be cleaner and more professional. As an initial dataset paper, the presentation could be polished further.
>
> We appreciate the reviewer's feedback regarding the presentation and formatting. In the final version of the paper, we will prioritize refining the figures and ensuring that the flavor annotation definitions are presented with clarity and professionalism. Our goal is to deliver a polished and cohesive manuscript for the camera-ready submission.
>
> >The dataset currently focuses only on wine in Western Culture. Expanding to more drink types would increase culture diversity.
>
> Good suggestion. The dataset has more wines from Western cultures, mainly because the Vivino platform is mostly used here. Expanding to include various drink types from other cultures is very interesting. We have added a discussion on the geo-cultural biases in the data.
>
> >The potential biases in the Vivino reviews (amateur enthusiasts) could be briefly discussed.
>
> See above

---

### Official Review · Reviewer_PNh1 · 2023-07-19
**Learning to Taste: A Multimodal Wine Dataset**

**Rating:** 5
**Confidence:** 4
**Correctness:** The claims mentioned in the submissio…

**Strengths:**

1. Novelty: The study introduces a unique, large, and multimodal dataset that includes human flavor annotations. This is a significant contribution as there are few datasets currently available that incorporate the sense of taste, especially in the field of wine tasting.
2. Multimodal Approach: The use of both text and images as input is an innovative approach that allows the model to make sense of more complex, multidimensional data, providing a richer and more detailed understanding of the wines.
3. Human-annotated Flavor Profiles: The use of human-annotated flavor profiles gives the model a more nuanced understanding of wine flavors. These human inputs provide a unique perspective that cannot be captured through traditional computational methods.
4. FEAST: The development of FEAST, which aligns the image and/or text embeddings with the human perception of flavor, represents an exciting advancement in the field of artificial intelligence, showing potential for applications in other areas beyond wine tasting.
5. Integration of Machine Learning and Food Science: By bridging the gap between machine learning and food science, the study opens up new avenues for interdisciplinary research, promoting a more comprehensive understanding of sensory inputs.
6. Extensive Dataset: With over 350k unique vintages, 897k images of wine labels, and 824k reviews of wines, the dataset is notably extensive. This depth and breadth of data give researchers ample material to train and test models, potentially leading to more accurate and reliable results.


**Additional Feedback:**

No additional feedback. I've already mentioned everything.

**Clarity:**

The paper provides a clear, well-structured, and insightful exploration of a novel area in flavor profiling using a multimodal approach. Despite some areas where the methodology could be advanced, the paper is commendably written, making it both accessible and understandable. Its contributions, coupled with its well-articulated structure, make it a valuable addition to the field of artificial intelligence applications in food science.

**Documentation:**

The dataset URL is provided.

**Ethics:**

No there are no ethical concerns with the submission.

**Limitations:**

Authors should take a look at the following comments and try to answer to it:

1. No Graph Models: Despite the inherent interconnectedness and potential for graphical representation in the data (e.g., wines as nodes, similarity in taste as edges), the study does not make use of graph models. Graph models could potentially provide a more accurate representation of the data's complex, non-linear relationships.
2. Age and Cultural Bias: The majority of participants in the wine-tasting events were between 21-25 years old and more than half were from Denmark. This could introduce a bias in the flavor annotations, which may not represent a wider range of cultural or age-related taste preferences.
3. Reliance on Human Input: While the use of human-annotated flavor profiles provides a unique perspective, it also means that the dataset is susceptible to the subjective biases of the annotators.
4. Subjectivity: The use of non-expert wine drinkers for annotating flavors introduces an element of subjectivity into the data. While this approach makes the study more accessible, it may compromise the accuracy or consistency of the flavor annotations.
5. Dependence on Pretrained Networks: The methodology relies heavily on pretrained networks like CLIP. This dependency could be viewed as a limitation, especially in situations where these pretrained models may not be applicable or available.

**Opportunities For Improvement:**

1. Development of New Models: The creation and use of new models, tailored specifically for interpreting and processing multimodal data and human sensory perception, could significantly enhance the ability of AI systems to understand complex sensory experiences like flavor. For example, models could be designed to better capture the subtleties and intricacies of flavor profiles, or to more effectively learn from the combination of textual reviews, images, and direct human input. These models could also include components that account for the influence of individual taste preferences, offering a more personalized understanding of flavor perception.
2. Use of Graph Models: Given the interconnected nature of the data, introducing graph models could offer a more accurate and detailed understanding of the dataset. Graph models could, for instance, represent wines as nodes and similarity in taste as edges, mapping out a complex network of flavor profiles. This could not only aid in understanding the relationships between different wines but also help uncover potential patterns or trends in the data that might otherwise go unnoticed with traditional models. Moreover, graph neural networks could be employed to better learn and generalize from the data.
3. Broadening Participant Demographics: The study’s data could benefit from a more diverse demographic of tasters. Currently, the majority of participants are young and mostly from Denmark, which might bias the flavor annotations. By incorporating a broader demographic spectrum, including varying ages, cultural backgrounds, and levels of wine expertise, the study could capture a more comprehensive understanding of wine flavor perceptions. This could lead to a more inclusive and generalizable model of flavor perception that applies to a wider audience.
4. Incorporating Objective Measures: While the study already includes human flavor annotations, which offer unique insights, it could be further enhanced by the inclusion of more objective measures. For instance, incorporating scientific analyses of the wines, such as chemical flavor profiles obtained through techniques like gas chromatography or mass spectrometry, could provide an additional layer of information about the wines. This could help counterbalance the subjective nature of human input, improving the robustness and validity of the study's findings. Furthermore, machine learning models could be designed or trained to interpret these chemical profiles and associate them with specific sensory experiences, paving the way for more accurate prediction and understanding of flavor perceptions.






**Relation To Prior Work:**

Based on the information provided, the paper appears to distinguish its work from previous studies primarily through its novel multimodal dataset and the introduction of the FEAST. It presents a unique combination of images, user reviews, and human-annotated flavor profiles, which appears to be a first in the realm of wine tasting. The FEAST methodology also seems to be a novel contribution that aligns the image and/or text embeddings with the human perception of flavor.

**Summary And Contributions:**

The paper "Learning to Taste: A Multimodal Wine Dataset" presents a unique contribution to multimodal data collection and flavor profiling in the realm of wine tasting. The study boasts considerable strengths, including the introduction of a novel, large-scale, and multimodal dataset encompassing human flavor annotations. This dataset, featuring over 350k unique vintages, 897k wine label images, and 824k wine reviews, significantly enriches the available data in this field.

The innovative application of both textual and visual data enhances the depth and nuance of the study, providing a more comprehensive perspective on the wines under examination. The use of human-annotated flavor profiles, meanwhile, offers a uniquely subjective angle that lends depth and individuality to the study's findings. Furthermore, the development and application of FEAST method demonstrates the promising potential for aligning complex data with the subtleties of human sensory perception.

Despite these strengths, the paper could benefit from certain improvements. Most notably, the absence of novel framework development and the lack of utilization of graph models, despite the clear interconnectedness of the data, are marked limitations. The study largely relies on existing methods like CLIP, t-STE, NMDS, CCA without proposing innovative algorithms or models. The paper's heavy reliance on pretrained networks like CLIP could also limit its applicability in situations where such models are not suitable or available.

The participants' demographics could also introduce a potential bias in flavor annotations as the majority were young and from a similar cultural background (Denmark). Additionally, the subjectivity inherent in using non-expert wine drinkers for flavor annotation may compromise the accuracy and consistency of the data.

In future research, developing new models tailored for multimodal data and human sensory perception could be a valuable step. It would also be beneficial to incorporate graph models to represent the complex relationships within the data better. Broadening participant demographics to better capture diverse taste preferences and incorporating more objective measures like chemical flavor profiles could counterbalance the subjective nature of human input, providing a richer and more comprehensive understanding of wine flavor.

In conclusion, while the paper makes a valuable contribution to the field of multimodal learning and flavor perception, its novelty is somewhat limited by its reliance on existing methods and models, and the absence of specific potentially beneficial techniques. Therefore, although the paper is an interesting and innovative piece of research, there are several areas where it could be further improved to increase its impact and applicability.

---

> ### Author Response · Authors · 2023-08-10
>
> >Development of New Models: [...]
>
> The development of novel, domain-specific models trained on multimodal data is certainly an exciting avenue of research. The main focus of the paper is to present a novel, multimodal dataset and benchmarking it using state of the art methods. We established a novel baseline that combines multiple modalities, and hope the community will improve upon it in future works!
>
> >Incorporating Objective Measures: [...]
>
> We agree that adding data from gas chromatography and mass spectrometry would be very interesting. In this paper, we have focused on human perception of flavor combined with text and image data. With this we have laid the foundation, such that other data modalities, can be incorporated later to further enhance our flavor understanding.
>
> >No Graph Models: [...]
>
> We agree with the reviewer’s suggestion that modeling the distances between wines on a graph has the potential to yield improvements upon the models benchmarked in the manuscript. The paper explores two methods to process the Napping data, based on previous work in sensory sciences and human kernel learning:
> 1. NMDS: We chose MDS because of the prevalent use of MDS in sensory science to process Napping data, as highlighted in lines 183-185 of our paper.
> 2. t-STE: This decision was inspired by related works in human kernel learning referenced in lines 97-99.
>
> We tried a graph representation where the wines are nodes and the similarity the edges as the reviewer suggested. We utilized the Node2Vec [https://snap.stanford.edu/node2vec/] method to reduce the graph and t-SNE to further reduce the graph to 2D. Here are our updated results:
>
> Human kernel | Acc [SVM]
> |-|-|
> | Random | 0.11
> | t-STE | 0.13
> | NMDS | **0.16**
> | Graph + node2vec + t-SNE | 0.15
>
> The table shows that using NMDS yields slightly better results than our graph representation. We think that further exploration of graph models to represent Napping data is a promising avenue of research — one which is made possible by the release of the dataset.
>
> >Age and Cultural Bias: [...]
>
> Your observation regarding age and cultural bias in our dataset is valid and well-taken. Indeed, our study did predominantly involve participants aged between 21-25, and a significant number were from Denmark.
> We recognize the importance of cultural contexts and personal experiences on flavor perception. Different cultural backgrounds can introduce varied taste preferences and perceptions, which might not be fully represented in our current dataset.
> Furthermore, age is another significant factor. As individuals grow older, their taste preferences and sensitivities might evolve. This dynamic nature of taste is something we've considered, and it's crucial to understand how age might play a role in the perception of wine flavors.
> We thank the reviewer for this comment, and we have added a paragraph to the discussion detailing the limitations imposed by the sampling bias of our cohort.
>
> >Reliance on Human Input: [...]
>
> We agree. The use of human-annotated flavor profiles provides a unique perspective! Data biases are not a unique problem to our approach. It is picked up by almost any deep learning model trained on human-generated data, e.g. google translate suggesting doctors are men, nurses women. It arises in image models that are racist. We agree this is a huge problem in the community and we hope that our dataset, and annotations, can provide a resource to study these biases. We have added a paragraph to the discussion in our manuscript describing the biases.
>
> >Subjectivity: [...]
>
> We reduce the sensitivity to the individuals’ taste by collecting at least five annotations per pairwise annotation. We further highlight some of the advantages of studying non-experts over experts:
> 1. **Generalizability:** Involving non-experts allowed us to capture a wider range of taste perceptions, reflecting the perspectives of the broader populace.
> 2. **Accessibility and Extensibility:** Engaging with non-experts renders our study both more approachable and more scalable.
> 3. **Wisdom of the crowd:** We believed that the sheer volume of data points collected would counterbalance the subjectivity.
> 4. **Commercial:** Collecting annotations from consumers has larger commercial value.
>
> We have added a paragraph to discuss the advantages of non-expert annotations.
>
> >Dependence on Pretrained Networks: [...]
>
> We believe that using pretrained models is an advantage! It shows that the method generalize and does not overfit to the specific dataset. Similar strategies are currently applied broadly in the field, where e.g. reliance on LLaMA[https://arxiv.org/abs/2302.13971] and StableDiffusion[https://openaccess.thecvf.com/content/CVPR2022/html/Rombach_High-Resolution_Image_Synthesis_With_Latent_Diffusion_Models_CVPR_2022_paper.html] are almost becoming de-facto. Could the reviewer elaborate on his/her/their concerns about applicability/availablity?

---

> > ### Comment · Reviewer_PNh1 · 2023-08-28
> >
> > Thank you for your clarifications in the rebuttal. While I appreciate your explanations and can see the rationale behind your choices, I would like to convey my stance on the matter.
> >
> > To address your specific query regarding pretrained models, I do recognize the advantage and widespread adoption of pretrained models in the community. As you pointed out, methods like LLaMA and StableDiffusion are indeed becoming prevalent and have shown their strength in various applications.
> >
> > However, my concern primarily lies in the fact that you did not fine-tune the pretrained model for your specific task or dataset. Pretrained models, while robust and general, often benefit from fine-tuning, as it allows them to adapt and specialize to the unique intricacies and nuances of the task at hand. By not fine-tuning, there's potential for leaving performance on the table, and it raises questions about the ultimate capability of your approach in its current form.
> >
> > That being said, I appreciate your effort and the discussions surrounding your paper. While I acknowledge the insights provided in the rebuttal, I remain of the opinion that the results could be further improved, and hence, I have decided to retain my original score.

---

> > > ### Author Response · Authors · 2023-08-28
> > > **Novelty of the paper**
> > >
> > > We thank the reviewer for elaborating on his/her stance.
> > >
> > > We emphasize that the novelty of this paper is **not** to obtain the highest possible accuracy on the dataset. Rather we focus on proposing a novel task (multimodal flavor understanding) and a novel large dataset that we benchmark with a rather simple out-of-the-box baseline. We agree with the reviewer that there are many ways to improve the performance, e.g. fine-tuning large models or using graph neural nets, etc. We hope this will excite many researchers in the machine learning and food science communities!
> > >
> > > Our results yield interesting conclusions, e.g. that combining multiple data modalities (flavor, image, and text) leads to improved performance. We hope the community will draw on these conclusions, use the provided data, and build models that improve upon the proposed baseline. We hope that the reviewer will consider basing his/her score on the focus of the paper (the proposed novel task of multimodal flavor understanding, the provided large dataset, and the interesting results on combining modalities) rather than providing evidence for well-established knowledge (ie. the ability to obtain slight performance gains with fine-tuning).

---

> > > > ### Comment · Reviewer_PNh1 · 2023-08-28
> > > >
> > > > Thank you for your response and the provided clarifications. I acknowledge the novelty surrounding the dataset and the combination of different sources of data, which is indeed of value to the research community.
> > > >
> > > > However, I still maintain my concerns regarding the model's performance. While I understand the intent behind using a simpler, out-of-the-box baseline model to offer an initial benchmark for the novel dataset and task, I believe the paper would have benefited significantly from more rigorous model experimentation. By presenting a stronger model as a benchmark, future research endeavors could have had a clearer path for improvement and innovation.
> > > >
> > > > I respect the aim of demonstrating the potential of the dataset and the new task, but I'm of the opinion that a part of demonstrating this potential involves showcasing at least a reasonably strong model performance.
> > > >
> > > > While the dataset's introduction and the combination of different data sources are commendable, the model's weak performance might overshadow these contributions in the eyes of some readers.
> > > >
> > > > Nevertheless, I hope my feedback will be seen as constructive criticism and will be considered in potential future works or revisions. The field needs innovative tasks and datasets, and I appreciate the effort put into introducing one.

---

### Official Review · Reviewer_1rM1 · 2023-07-21
**The paper is interesting, while some parts should be**

**Rating:** 6
**Confidence:** 3
**Correctness:** 1. The accuracy in Tab. 1, 2, 3 in ex…
**Clarity:** 1. In line 35, why five wines yield 1…

**Strengths:**

1. The work is novel for the flavor modality.
2. The work is easy to follow.
3. The experiments are comprehensive.

**Additional Feedback:**

No

**Documentation:**

Yes, its section 3 gives a detailed introduction of crawl process.

**Ethics:**

Yes, I wonder whether the work is allowed to crawl the images, reviews and tags from the website. The license should be discussed.

**Limitations:**

When release the dataset, whether the license of the multi-modal data source is obeyed?

**Opportunities For Improvement:**

1. When we can have a better representation to improve the flavor modality?
2. There are still some points need to be explained. Please check "Clarity" section.
3. The experimental task setting should be clearly defined.

**Relation To Prior Work:**

The work is related to multimodal representations, quantifying flavor, and human kernel learning. All these parts has been well discussed.

**Summary And Contributions:**

This work builds a new modality flavor, which is novel. To collect this modality data, this work use the wine to collect. Based on the flavor modality, this work further collects the wine images, wine reviews, and wine attribute tags. Together the four modality, this work creates a new dataset, called WineSensed. It consist of 897k images of wine labels and 824k reviews of wines as well as 350k vintages.

---

> ### Author Response · Authors · 2023-08-10
>
> >When can we have a better representation to improve the flavor modality?
>
> The current multimodal data have been shown to provide strong flavor embeddings. We believe that together with the community we can develop better algorithms for the latent-space representation of flavor. While there are certainly low-hanging fruit on the analysis front, there is also the possibility of enriching the dataset with more modalities, such as chemical compositions. However, those are difficult data to obtain en masse, and we believe the dataset contains enough signal to yield accurate latent representations of wines as improvements in the integration of data modalities continue to compound.
>
> >The experimental task setting should be clearly defined
>
> We have detailed the experimental task setting in the Appendix under the sections "Details for coarse-grained flavor predictions" and "Details for fine-grained flavor predictions." We will make the references to these sections more clear.
>
> >When release the dataset, whether the licence of the multi-modal data source is obeyed?
>
> See answer below
>
> >The accuracy in Tab. 1, 2, 3 in experimental setting is not clear to me. Please give a clear definition of the task setting regarding the accuracy.
>
> Thank you for bringing this to our attention. In Tables 1, 2, and 3, the accuracy figures represent the average classification accuracy across all wine attributes. These attributes, which include country, grape, region, price, rating, year, and alcohol content, are detailed in Figure 5.
>
> >In line 35, why five wines yield 10 pairwise?
>
> In line 35, when you consider pairwise distances between five wines (A, B, C, D, and E), you're essentially determining how each wine compares with another. Here's the breakdown:
>
> 1. Wine A is compared with wines B, C, D, and E: giving 4 comparisons.
> 2. Wine B is then compared with wines C, D, and E: adding 3 more comparisons.
> 3. Wine C is compared with wines D and E: 2 more comparisons.
> 4. Lastly, wine D is compared with wine E: 1 more comparison.
>
> Adding all these together, you get 4 + 3 + 2 + 1 = 10 pairwise comparisons.
>
> >Since the work uses pairwise distance to represent the flavor modalities, whether the flavor modality is sparse and how do you address sparse problem?
>
> Good question. In processing our human-annotated data, we adopted two main methodologies: the creation of a distance matrix and the formulation of triplets.
>
> 1. Distance Matrix: We calculated the Euclidean distances between each pair of annotated wines, compiling these into a matrix. Where there were unobserved or missing annotations between wines, the cell in the matrix was set to zero. The NMDS approach we utilized is adept at handling scenarios where not all dissimilarities are observed. For instance, if we only have a fraction of the dissimilarities, NMDS accommodates for these missing observations by altering the stress definition. By setting the "metric" parameter to "False" in NMDS, it acknowledges zeros as indicators of unobserved dissimilarities, handling missing data without compromising the integrity of the representation.
>
> 2. Triplets Creation: We also leveraged the Euclidean distances to formulate triplets. These triplets were intended to convey relative similarities, such as wine "i" being more similar to wine "j" than wine "k". By relying on relative distances instead of absolute ones, t-STE sidesteps the complications associated with dense pairwise distance matrices.
>
> During our human-annotation events, we ensured that every participant received a random combination of five wines to taste, and that each combination was unique. By doing this, the chances of having dense clusters of known values in one section of the matrix and a sparse cluster in another are reduced. Instead, the observed values (non-zero entries) are more evenly distributed across the matrix.
>
> >In line 141, how many different colors are used for colored stickers?
>
> The number of distinct colors used for the stickers varied. During each phase of the human annotation experiment, we used between 10-15 different colors, contingent upon the number of bottles open. That said, each napping paper featured only five unique colors, since participants were comparing and annotating similarities among five wine samples in each round.
>
> >I wonder whether the work is allowed to crawl the images, reviews and tags from the website. The licence should be discussed.
>
> Regarding concerns about the ethical collection of images, reviews, and tags from the website: In this instance, the data utilized for our project was obtained through a collaborative partnership with Vivino. Our co-author, Alireza Kashani, who is an employee at Vivino, facilitated the provision of this data. Thus, all information was shared with the appropriate permissions and in line with the platform's terms and conditions. The dataset is published under the CC BY-NC-ND 4.0 Licence, as shown on the project webpage.

---

> ### Comment · Reviewer_1rM1 · 2023-08-22
>
> Thanks a lot for the authors' detailed explanation.
>
> My concern has been mainly addressed.
>
> So I keep my original score.

---

### Official Review · Reviewer_YEVG · 2023-07-21
**See summary and contributions**

**Rating:** 6
**Confidence:** 5

**Strengths:**

The authors explore an uncharted direction for multimodal representations by combining images, text and human perception of flavor. This novelty opens the possibility for new studies by designing original experiments leveraging the peculiar nature of the dataset. I particularly appreciated how the authors proved that the ordering of distances in the latent space corresponds to the human perception of flavor.


**Additional Feedback:**

None

**Clarity:**

The paper has an engaging writing style following an enough clear and logical structure.


**Correctness:**

The claims made in the submission are correct, if one ignored the improvements. The dataset is constructed in a sound way, the human- annotation of flavors has been carried out with particular care. The evaluation methods and experiment design are appropriate, however they have a big room for improvement as discussed in “Limitations”.


**Documentation:**

The dataset is sufficiently documented. The website and github page ensure reproducibility and provide enough information

**Ethics:**

The submission doesn’t generate any ethical concerns.

**Limitations:**

I find important to extend the work to other foods and beverages, however the authors didn’t reflect enough on what are the limitations and future directions of the solution, providing an unsatisfactory discussion. I suggest improving the coarse flavor predictions by replacing the SVM classifier with a neural network, so that the pretrained models can be fine-tuned on the classification task. I also suggest trying larger language models (e.g. BART-large) and better performing ones (e.g. FLAN-T5, PEGASUS).


**Opportunities For Improvement:**

The experimental part of the work has room for improvement. In particular the coarse flavor predictions provide unsatisfactory results, given that they don’t deviate much from the random baseline. Additionally, working exclusively on wines might limit the impact on the research community.


**Relation To Prior Work:**

Despite the peculiar nature of the dataset, the authors managed to collect a sufficient amount of literature covering previous contributions. Related work is organized into sections and for each one of them the authors clearly argued how this work differs.


**Summary And Contributions:**

The paper is well written, and the dataset is appropriately constructed and documented. However, the experimental part conceals the real potential of the dataset and needs to be improved.
"WineSensed" is a large multimodal wine dataset that combines 897k wine images and 824k reviews from the Vivino platform. It includes over 350k unique vintages with annotations like year, region, rating, alcohol percentage, price, and grape composition. The dataset also features fine-grained flavor annotations obtained through a wine-tasting experiment involving 256 participants. The authors also introduce a low-dimensional concept embedding algorithm that integrates human experience with automatic machine similarity kernels. This dataset offers valuable insights into visual perception, language, and flavor relations in wines.

---

> ### Author Response · Authors · 2023-08-10
>
> >The experimental part of the work has room for improvement. In particular the coarse flavor predictions provide unsatisfactory results, given that they don’t deviate much from the random baseline. Additionally, working exclusively on wines might limit the impact on the research community.
>
> The random baseline has 11 % accuracy, whereas best model presented in the paper has 28 % accuracy. We think this is a significant improvement, especially given that the models are not training on the specific task at hand. However, our main insight is that using multiple modalities improves performance, e.g. 3 modalities > 2 modalities > 1 modality. We believe this is a very interesting finding that might guide more sophisticated methods to beat our baseline.
> It is not uncommon for benchmarks published along with novel datasets to have low accuracy. Compare for instance the 17.3% (vs. 1/200 random) benchmarked accuracy of the CUB200 dataset upon release (https://authors.library.caltech.edu/27452/1/CUB_200_2011.pdf) to the 93% accuracy achievable on the same dataset today (https://paperswithcode.com/sota/fine-grained-image-classification-on-cub-200). Part of the excitement of a novel dataset is that there is usually room for improvement on the analysis and modeling front.
> We have tried to replace the SVM with a small MLP, and explored FLAN-T5, PEGASUS, BART-large. Here are the updated tables:
>
> | Machine kernel | Modality | Acc [SVM] | Acc [NN]
> |------------|-----------|-----------|-----------|
> | Random |  | 0.11 | 0.11 |
> | ViT      | Image | 0.09 | 0.13 |
> | DeiT       | Image | 0.14 | 0.15 |
> | ResNET | Image | 0.15 | 0.16 |
> | CLIP | Image | 0.11 | 0.15 |
> | T5 | Text | 0.15 | 0.16 |
> | ALBERT | Text | 0.15 | **0.18** |
> | BART | Text | **0.16** | 0.15 |
> | DistilBERT | Text | 0.15 | 0.17 |
> | CLIP | Text | **0.16** | **0.18** |
> | FLAN-T5 | Text | 0.15 | 0.17 |
> | PEGASUS | Text | 0.13 | 0.13 |
> | BART-large | Text | 0.11 | 0.15 |
>
> Modality | Acc [SVM] | Acc [NN]
> |------------|-----------|-----------|
> | Flavor | 0.16 | 0.11 |
> | Image | 0.11 | 0.15 |
> | Text | 0.16 | 0.18 |
> | Text+Flavor | 0.23 | 0.18 |
> | Image+Text | 0.22 | 0.25 |
> | Image+Flavor | 0.23 | 0.18 |
> | Image+Text+Flavor | **0.28** | **0.26** |
>
> Human kernel | Acc [SVM] | Acc [NN]
> |------------|-----------|-----------|
> | Random | 0.11 | 0.11 |
> | t-STE | 0.13 | 0.10 |
> | NMDS | **0.16** | **0.11** |
>
> Reducer | Acc [SVM] | Acc [NN]
> |------------|-----------|-----------|
> | UMAP | 0.15 | 0.18 |
> | PCA | 0.20 | 0.21 |
> | t-SNE | **0.22** | **0.25** |
>
> Combiner | Acc [SVM] | Acc [NN]
> |------------|-----------|-----------|
> | ICP | 0.21 | 0.24 |
> | SNaCK | 0.23 | 0.24 |
> | CCA | **0.28** | **0.26** |
>
> We find that generally, the neural network does not increase classification performance. We believe that the SVM performs better because it is more robust to unbalanced classes. Overall conclusion remains the same - using multiple modalities significantly improves the performance. We will update the paper accordingly.
> Furthermore, while the exclusive focus on wines in our research does present certain limitations in terms of broader impact, we believe our study serves as a valuable foundation for subsequent investigations. Our intent is that these methods and findings can be extended to a wider array of food and beverage categories, thereby potentially increasing their relevance and influence within the research community.
>
> >I find important to extend the work to other foods and beverages, however the authors didn’t reflect enough on what are the limitations and future directions of the solution, providing an unsatisfactory discussion. I suggest improving the coarse flavor predictions by replacing the SVM classifier with a neural network, so that the pretrained models can be fine-tuned on the classification task. I also suggest trying larger language models (e.g. BART-large) and better performing ones (e.g. FLAN-T5, PEGASUS).
>
> See above

---

### Official Review · Reviewer_DdVt · 2023-07-26
**Good contribution at the intersection of Machine Learning and Food Science.**

**Rating:** 10
**Confidence:** 4

**Strengths:**

- Extensive dataset
- Introduces FEAST, a way to augment food metadata with subjective taste rankings by aligning the embedding spaces
- Includes mutliple ablations to understand the effect of the different modalities, and finds that the taste embeddings are useful
- The data is very well documented
- The code to reproduce the study is well documented and the study can be extended to other food datasets.

**Additional Feedback:**

- Please cite the packages that you use. For example, `transformers` and `pandas` both have official bibtex entries, and I'm sure other packages do.
- It might be interesting to include an actual image (the ones from the website) of the digitization process described in lines 145-150. This would also help elucidate the Napping method.

- The vertical line in Table 1 can probably be left out.

- Can the data be used to see whether one can detect differences between "New World" and "Old World" wines?

- It would be interesting to include a text analysis of the reviews.

**Clarity:**

The paper is extremely well written (some very minor spelling error that might warrant an additional editing pass), easy to follow, and appropriately detailed.

**Correctness:**

Some questions that arose:

- Why was the alignment of the two modalities conducted in 2D space? Line 178 claims it to be simpler, but wouldn't it be more correct to first align and then reduce dimensions?

- Why wasn't Procrustes considered for aligining

**Documentation:**

Data and data-adjacent processes are all meticulously documented and described in detail, both in the paper and in the appendices (PDF and website).

**Limitations:**

- The paper does not mention whether the data source biases (most wines being Italian e.g.) can affect the analysis.

- Can the experience / background of the participants (line 134) affect the taste embeddings, or is Napping robust to that?

**Opportunities For Improvement:**

See other sections.

**Relation To Prior Work:**

The paper does a great job in carving out a detailed image of the field and positioning itself within it.

**Summary And Contributions:**

This paper introduces a multimodal wine dataset. The first constituent part of the dataset is sourced from the vivino website, and each row corresponds to a vintage and includes: multiple user-submitted photos, multiple user reviews, and vintage metadata (region, grape, alcohol, etc.). The second constituent part of the dataset is a human-generated pairwise similartiy ranking of a subset of vintages, based on the *Napping* method.

The paper further demonstrates how to convert these pairwise judgments to flavor embeddings, how to align those flavor embeddings with those of the other modalities (text and image) to enable both coarse and fine-grained analyses of vintages using subjective human judgments, effectively augmenting the first constituent part of the dataset with crowdsourced flavor information.

---

> ### Author Response · Authors · 2023-08-10
>
> >The paper does not mention whether the data source biases (most wines being Italian) can affect the analysis
>
> We have added a paragraph in the discussion detailing the limitations due to the majority of the human annotations being on Italian wines. The advantage of the dominance of Italian wines is that it leads to a more nuanced distinction among them due to the narrower scope of origin, e.g. modelling more intricate flavor nuances.
>
> >Can the experience / background of the participants (line 134) affect the taste embeddings, or is Napping robust to that?
>
> Indeed, the prior experiences and personal backgrounds of participants can exert influence on sensory evaluations, to a certain extent. Taste perception is inherently subjective and may be swayed by a myriad of factors such as age, cultural upbringing, dietary preferences, and past exposure to various tastes or flavors (https://www.sciencedirect.com/science/article/abs/pii/S0195666381800062, https://www.ncbi.nlm.nih.gov/pmc/articles/PMC2364714/, https://flavourjournal.biomedcentral.com/articles/10.1186/2044-7248-4-9). This underlines the role that individual backgrounds and experiences can play in shaping perceptions of taste, aroma, and other sensory attributes.
>
> Despite the comparative emphasis of the Napping method, it is not entirely successful in eliminating the influences of individual variance. The potential for personal biases persisting in the data cannot be completely discounted, even with this safeguard in place. We limit it by recording each pairwise distance at least five times. We will include a discussion on this.
>
> >Why was the alignment of the two modalities conducted in 2D space? Line 178 claims it to be simpler, but wouldn’t it be more correct to first align and then reduce dimensions?
>
> The alignment of the modalities was conducted in a 2D space primarily to facilitate easier and more effective visualization. This approach allowed for a more intuitive understanding of the data alignment and structure during the analysis process. Furthermore, the comparison of the methods is also simpler, when they all use the same dimensionality.
>
> >Why wasn’t Procrustes considered for aligning?
>
> We tried Procrustes as a combination method. Here are our updated results:
>
> | Combiner   | Acc [SVM] |
> |------------|-----------|
> | ICP        | 0.21 |
> | SNaCK      | 0.23 |
> | CCA        | **0.28** |
> | Procrustes | 0.19 |
>
> Our overall conclusion remains the same. We will update the paper accordingly.
>
> >Minor spelling mistakes are present in the paper
>
> Thank you for notifying us, we have updated the manuscript to fix spelling errors.
>
> >Please cite the packages that you use. For example, transformers and pandas both have official bibtex entries, and I'm sure other packages do.
>
> Thank you for catching this. The section "Implementation details for flavor space generation" in the Appendix cites the packages mentioned above. We have added a citation to this section in the Experiments section of the paper for visibility.
>
> >It might be interesting to include an actual image (the ones from the website) of the digitization process described in lines 145-150. This would also help elucidate the Napping method.
>
> We thank the reviewer for the suggestion, we have added the images of the napping papers from our website to the Appendix of the manuscript and referenced them in lines 145-150.
>
> >The vertical line in Table 1 can probably be left out.
>
> We thank the reviewer for the note, we have removed the vertical line from Table 1 in the paper.
>
> >Can the data be used to see whether one can detect differences between "New World" and "Old World" wines?
>
> Interesting question! Our dataset encompasses a significant amount of data on wines from both of these categories. In figure 4 a) you can see the distribution of wines by country, where you can see the dataset has wines from the "Old World": Italy, Spain, France and Portugal, as well as the "New World": Argentina, Australia, United States and South Africa. Given this diversity, it would certainly be intriguing to conduct an analysis to identify any distinct clusters or patterns that might help differentiate between these two categories. One of the privileges of collecting our dataset is how many interesting questions such as these arose along the way, to the point where we had to be selective of the benchmarks presented in the manuscript. The data is available for academic research, which makes it possible for anyone whose curiosity is piqued, to conduct such a study.
>
> >It would be interesting to include a text analysis of the reviews.
>
> We present summary statistics on the reviews in Figures 4 b) and 4 c). There are many interesting further analysis that can be done with the reviews, including sentiment analysis, finding prototypical review, summarization, with more. We believe many of these are out-side the scope of this paper. However, we are curious specifically what type of text analysis the reviewer imagined?

---

> > ### Comment · Reviewer_DdVt · 2023-08-31
> >
> > Dear authors,
> >
> > Thank you for addressing my comments.
> >
> > While I disagree with your choice of aligning in 2D and remain unconvinced with your answer, I have come to realize that this is a minor issue from the vantage point of the D&B track itself; I have therefore upgraded my score.
> >
> > I agree that a deep text analysis might be rather out of scope, but I had something like an LDA topic model in mind. (Figure 4c is a bit tough to read, both because of the size of the text and its orientation. It would be nice if you could improve that).

---

### Author Response · Authors · 2023-08-10
**Summary of Reviewers Positive Remarks**

We thank all reviewers for their positive and constructive feedback. Our dataset was appreciated for its "significant contribution as there are few datasets currently available that incorporate the sense of taste" [R4]. Highlighting its depth, one reviewer noted "With over 350k unique vintages, 897k images of wine labels, and 824k reviews of wines, the dataset is notably extensive" [R4]. “The authors explore an uncharted direction for multimodal representations by combining images, text and human perception of flavor. ”[R1] Another emphasized its importance stating, "The dataset is unique and highly relevant, as flavor is an important but (extremely) understudied modality for multimodal deep learning" [R5], and "The large scale of the study (256 people, 5K annotations) is impressive" [R5]. The experiments are "comprehensive" [R3] and “... cover many encoder, dimensionality reduction, and fusion options. The gains from multimodal learning validate the utility of the dataset.” [R5] The development of FEAST described as "an exciting advancement in the field of artificial intelligence, showing potential for applications in other areas beyond wine tasting" [R4]. On the presentation of our work, the paper was praised for being "extremely well written" [R1], “easy to follow” [R3], and having "an engaging writing style following a clear and logical structure" [R2]. Furthermore, its unique contributions and structure were commended to "make it a valuable addition to the field of artificial intelligence applications in food science" [R4]. In the following, we will address the reviewer's comments.

---

### Decision · Program_Chairs · 2023-09-22

**Decision:**

Accept (Poster)

**Comment:**

This paper has received the scores of 5, 6, 6, 6, and 10, where all reviewers unanimously voted for accepting the paper after reading the authors’ rebuttal.

Most of the reviewers appreciated the importance of such an innovative task and benchmark, but there still remains a concerns about the baseline models’ performances.
In this respect, I recommend accepting the paper as a poster.